# Spatiotemporal control of signal-driven enzymatic reaction in artificial cell-like polymersomes

Hanjin Seo[1] & Hyomin Lee �ORCID [1] ✉

Living cells can spatiotemporally control biochemical reactions to dynamically assemble membraneless organelles and remodel cytoskeleton. Herein, we present a microfluidic approach to prepare semi-permeable polymersomes comprising of amphiphilic triblock copolymer to achieve external signal-driven complex coacervation as well as biophysical reconstitution of cytoskeleton within the polymersomes. We also show that the microfluidic synthesis of polymersomes enables precise control over size, efficient encapsulation of enzymes as well as regulation of substrates without the use of biopores. Moreover, we demonstrate that the resulting triblock copolymer-based membrane in polymersomes is size-selective, allowing phosphoenol pyruvate to readily diffuse through the membrane and induce enzymatic reaction and successive coacervation or actin polymerization in the presence of pyruvate kinase and adenosine diphosphate inside the polymersomes. We envision that the Pluronic-based polymersomes presented in this work will shed light in the design of in vitro enzymatic reactions in artificial cell-like vesicles.

Cells carry out variety of complex biochemical reactions that are delicately orchestrated in both space and time. Specifically, the cell interior is compartmentalized by a membrane that separates and protects the vital cellular elements from the surrounding environment while performing central functions by regulating the internal composition[1,2]. While the cellular membrane is semi-permeable and selectively responds to changes in the surrounding environment[3], subcellular structures within the membrane including membraneless organelles (MOs) such as P bodies, Cajal bodies, and speckles provide additional means to spatially modulate the dynamic behavior of biochemical reactions within the cell[4–6]. These MOs based on liquid-liquid phase separation (LLPS) systems and complex coacervates have been shown to catalyze biochemical reactions[7], and provide a dynamic environment for the spatiotemporal coordination of reactions inside cellular structures due to high mass transfer rate and stimuli-responsiveness[8–12].

To mimic these MOs, various bottom-up approaches have been devised to prepare synthetic cellular models embedding imperative aspects of living cells[13,14]. One promising method to form these artificial cell models is the droplet microfluidics, in which precise control of fluids have been utilized to synthesize emulsion templated giant unilamellar vesicles (GUVs) that are comparable in size to an actual cell[7–9,11,12]. However, the modular design of in vitro biochemical reactions in GUVs requires precisely tunable membrane property including thickness, surface potential, bending modulus, and most importantly, permeability, depending on the reaction of interest[3]. While the permeability depend on whether they consists of synthetic polymer or lipid, biopores such as α-hemolysin and melittin have been also inserted into these polymersome and liposome membranes to regulate material transport[6,15,16]. However, biopores are difficult to purify and insert desired quantity into the membrane, all of which limit the usage of these biopores in organizing complex reactions inside artificial cells. Moreover, while high molecular weight of block copolymers compared to lipids grant enhanced mechanical durability, the thickness of the resulting membrane is often too thick to effectively accommodate the membrane protein and the inherent diffusivity is low. In response to these demands, polymersomes with various compositions have been explored as the type and molecular weight of each

[1]Department of Chemical Engineering, Pohang University of Science and Technology (POSTECH), Pohang 37673, Republic of Korea.
✉e-mail: hyomin@postech.ac.kr

block in the amphiphilic block copolymer can be freely adjusted to control the permeability[6,17,18].

Integrating the interconnectivity of consecutive biochemical reactions in artificial cells provides a bottom-up approach to design responsive synthetic cellular models[16,19]. For instance, based on the motif that cells require continuous feeding of materials and energy for their vitality through active metabolism based on biocatalysts, enzymatic reactions have been tailored in GUVs for emulating the cellular function[20–23]. Moreover, light-triggered synthesis of ATP via ATP synthase, followed by actin polymerization has been recently exploited in lipid-based vesicles to reconstitute photosynthetic organelles in protocellular systems[24]. However, elaborate machinery of external signal-driven enzymatic reaction or cascade reaction in artificial cells still remains unclear as the membrane permeability, stoichiometry of the reactants, and reaction kinetics, all needs to be carefully considered. Therefore, there is an unmet need for a new synthetic cellular model with in-depth understanding of the molecular transport property for successful implementation of enzymatic reactions.

Herein, we introduce a microfluidic approach to achieve spatio-temporal control of enzymatic reactions in artificial cell-like polymersomes without usage of biopores. The microfluidic synthesis of polymersomes enables precise control of vesicle size, efficient encapsulation of biomaterials with minimal damage to enzymes. In addition, the resulting polymersomes comprising of poly(ethylene glycol)-b-poly(propylene glycol)-b-poly(ethylene glycol)(Pluronic L121) are stable and semi-permeable, thereby offering controlled molecular transport across the membrane. In addition, we show that the membrane is size-selective and permeable to protons and small molecules by demonstrating pH-induced coacervate formation

between nucleotide and polyamines and signal-driven enzymatic reaction in polymersomes. As these polymersomes allow small external signals to readily diffuse through the membrane, infusing phosphoenol pyruvate (PEP) in the vicinity of polymersomes induces enzymatic reaction which leads to nucleation and maturing of complex coacervate droplets inside the polymersomes. Furthermore, we also extend this concept to cascade reaction and successive polymerization of globular actin monomers into actin filaments for regulation of cytoskeletons.

# Results

## Microfluidic fabrication and characterization of Pluronic-based polymersomes

To prepare Pluronic-based polymersomes, we produce water-in-oil-in-water ($W_1/O/W_2$) double emulsion droplets using a glass capillary-based microfluidic device (Fig. 1a, Supplementary Fig. 1). During droplet formation, the inner aqueous phase ($W_1$) is injected through the smaller cylindrical capillary while the middle oil phase (O), a mixture of chloroform and cyclohexane (36:64 vol%) containing 20 wt% Pluronic L121 is injected through the injection capillary. Additional aqueous continuous phase ($W_2$) is injected from the same side through the interstices of the square and injection capillary. The stream of water-in-oil ($W_1/O$) droplets from the injection capillary breaks up into monodisperse double emulsion droplets with ultra-thin shells by shearing of the aqueous continuous phase (Supplementary Movie 1). Upon collection of these emulsion droplets, chloroform readily evaporates leading to thinning of the oil shell[25,26], followed by dewetting of the cyclohexane droplet within few minutes to form a polymer bilayer comprising of Pluronic L121[27,28], as shown in Fig. 1b and Supplementary

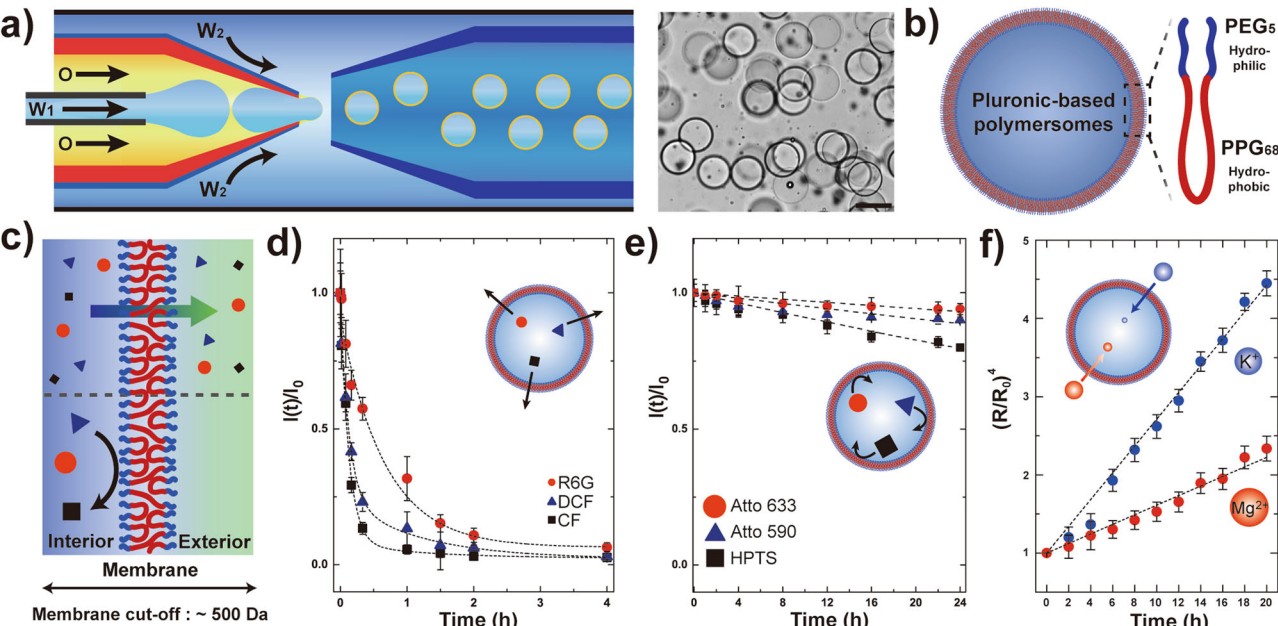

**Fig. 1 | Microfluidic fabrication of semi-permeable polymersomes. a** Schematic illustration and optical micrograph showing the production of Pluronic-based polymersomes via glass capillary-based microfluidic device using water-in-oil-in-water ($W_1/O/W_2$) double emulsion droplets as templates. Scale bar represents 100 μm. **b** Schematic illustrations of the produced polymersomes consisting of Pluronic L121. PEG: poly(ethylene glycol), PPG: poly(propylene glycol). **c** Schematics illustrating the experiment used to determine the membrane permeability towards fluorescent dye molecules with different molecular weight and charge. Fluorescent dye molecules with molecular weight <500 Da readily diffuse through the membrane (top) while the larger ones do not (bottom). **d** Plot showing the normalized fluorescence intensity change within the polymersomes over time for three representative fluorescent dye molecules smaller than the MW cut-off of the membrane. Small cationic dye (red circle) refers to Rhodamine 6 G

(R6G, 479 Da), while the neutral (blue triangle) and anionic ones (black square) are 2,7-dichlorofluorescein (DCF, 401 Da) and carboxyfluorescein (CF, 376 Da), respectively. **e** Plot showing the normalized fluorescence intensity change within the polymersomes over time for three representative fluorescent dye molecules larger than the MW cut-off of the membrane. Large cationic dye (red circle) refers to Atto 633 NHS ester (Atto 633, 749 Da), while the neutral (blue triangle) and anionic ones (black square) are Atto 590 (691 Da) and 8-hydroxypyrene-1,3,6-trisulfonic acid (HPTS, 524 Da), respectively. The dotted lines in both plots represent the model fit used to determine the membrane permeability for each diffusing species. **f** Plot showing the polymersomes swelling with respect to time ($n = 11$) which is used to acquire the membrane permeability towards two metal ions, potassium ($K^+$, blue) and magnesium ($Mg^{2+}$, red), respectively. All error bars represent standard deviation, in **d–f**.

Fig. 2. We note that minuscule oil pockets remain attached to the polymersomes but eventually thins out over time and do not affect the experimental results. Unlike the conventional lipid-based giant uni-lamellar vesicles, the Pluronic-based polymersomes are stable for at least 1 week at room temperature as well as 37 °C (Supplementary Fig. 3 and 4) and monodisperse in size with mean diameter of 118 μm and coefficient of variation (CV) of 4.0% (Supplementary Fig. 5). Also, to determine the permeability of the membrane, we prepare sets of polymersomes each encapsulating six representative fluorescent dyes with different molecular weight (MW) and net charge in the aqueous core and monitor the retention of these dyes over time. We note that the polymersome interior and exterior are osmotically matched, $100 \pm 3$ and $103 \pm 4$ mOsm/kg, respectively, and fluorescent dye molecules are added in low concentrations (5 ppm) to minimize this difference. While numerous factors are known to affect the physical properties of the polymeric membrane[29,30], we anticipated that the polymersomes comprising of Pluronic L121 with low hydrophilic-to-hydrophobic volume fraction ($f < 0.18$) and hydrophilic-to-lipophilic balance (HLB ~ 7) result in a less densely structured membrane than analogous polymersomes with higher f values and therefore would be permeable to molecules smaller than the MW cut-off while the larger ones are not[27,28], as depicted in Fig. 1c. Indeed, monitoring the nor-malized fluorescence intensity change within the polymersomes over time reveals that the release profile depends on both the MW and the net charge as shown in Fig. 1d, e. We also observe abrupt difference in the release profiles for sets of fluorescent dye molecules with MW >500 Da compared to the smaller analogs, indicating that the MW cut-off the membrane can be estimated as 500 Da.

To further confirm this, we estimate the permeability by adapting the simplified solution of Fick's law[31], and fitting the normalized fluorescence intensity change acquired from Fig. 1d, e as detailed in the Methods section. We observe a clear tendency that the permeability decreases from 33.6 ($P_{CF}$), 26.1 ($P_{DCF}$), 6.28 ($P_{R6G}$), 0.051 ($P_{HPTS}$), 0.026 ($P_{Atto590}$), to 0.016 nm s$^{-1}$ ($P_{Atto633}$) with increase in MW of the diffusing species (Supplementary Fig. 6). These results confirm that the esti-mated MW cut-off value of 500 Da is valid[32–35], as evidenced by the almost two orders less in permeability value from R6G (479 Da, $P_{R6G} = 6.28$ nm s$^{-1}$) to HPTS (524 Da, $P_{HPTS} = 0.051$ nm s$^{-1}$).

To verify whether the Pluronic-based polymersome membrane is also permeable to potassium (K$^+$) and magnesium (Mg$^{2+}$) ions, we employ the method proposed by other in which volumetric swelling of polymersomes is monitored under osmotic pressure in the pre-sence of permeable ions to acquire the ion permeability ($P_{ion}$)[28]. As Pluronic-based polymersomes exhibit a MW cut-off value of 500 Da, encapsulation of 10 wt% PEG (MW 6000 Da, $100 \pm 3$ mOsm kg$^{-1}$) in polymersomes and subsequent exposure to hypertonic solutions comprising of ions smaller than the membrane MW cut-off allows determination of the permeability of K$^+$ and Mg$^{2+}$ through the membrane. Here, hypertonic solution comprising of either 3.72 wt% KCl or 3.12 wt% MgCl$_2$ solutions with osmolyte concentration of $888 \pm 5$ and $897 \pm 3$ mOsm kg$^{-1}$, respectively, were used as the hypertonic solutions. Acquisition and subsequent analysis of the polymersomes swelling data upon incubation in either hypertonic KCl or MgCl$_2$ solution reveals that K$^+$ exhibit higher permeability ($P_{K^+} = 0.65$ nm s$^{-1}$) than Mg$^{2+}$ ($P_{Mg^{2+}} = 0.23$ nm s$^{-1}$) (Fig. 1f), which is coherent with previous reports[36,37]. The estimated permeability values of K$^+$ and Mg$^{2+}$ ions are smaller than the fluorescent dye molecules, possibly due to the high electric charge density which leads to larger hydrodynamic radius[38,39]. The permeability of these two ions are separately confirmed with additional experiments in which K$^+$ permeability is compared before and after inclusion of K$^+$ ion-selective channel (valinomycin) and permeation of Mg$^{2+}$ is ver-ified by eriochrome black-T complexometric titration, as noted in Supplementary Fig. 7. Overall, these results indicate that the Pluronic-based polymersomes presented in this work exhibit a molecular cut-off of ~500 Da and are also permeable towards metal ions such as K$^+$ and Mg$^{2+}$ that are essential in various enzymatic reactions[40].

## pH-induced complex coacervation in polymersomes

The assembly of complex coacervates has been widely studied to mimic the sub-cellular structures such as MOs[41]. Prior to the assembly of model complex coacervates in polymersomes, we encapsulate 5 μg mL$^{-1}$ HPTS solution adjusted to pH 4 in polymersomes to verify whether the membrane is proton permeable. As HPTS exhibits absorption shift above pH 7.3, we utilized the pH sensitive nature of HPTS to determine the proton permeability. We find that when the pH was increased by adding pH 10 buffered solution to the periphery of polymersomes, the fluorescence intensity enhanced dramatically in <1 s due to rapid permeation of proton through the membrane fol-lowed by deprotonation of HPTS (Supplementary Fig. 8). While alter-ing pH does not provide the absolute value of diffusivity due to dilution effect[42], this simple visualization method reveals that proton rapidly diffuses through the membrane.

Next, we studied the coacervate formation between adenosine triphosphate (ATP) and poly(allyl amine hydrochloride)(PAH) in bulk (Fig. 2a). Here, ATP with MW (507 Da) larger than the membrane MW cut-off was selected as the model nucleotide, and thus can be effec-tively retained within the Pluronic-based polymersomes (Supplemen-tary Fig. 9). PAH was chosen as the representative positively charged polyelectrolyte due to non-toxicity, biocompatibility, and structure similarity to cellular polyamines, which are essential for normal cell growth and viability in eukaryotic cells[43]. More importantly, ATP and PAH serve as an excellent pH-responsive biomimetic MO model[44], since they each have distinctive pKa values of 6.5 and 9 respectively. Indeed, we observe no apparent coacervate formation between ATP and PAH at pH 4 and pH 11, which is either lower than the pKa of ATP or higher than the pKa of PAH. However, at the intermediate pH of 7.4 at which both ATP and PAH are oppositely charged, complex coacervate forms within a minute (Supplementary Fig. 10).

To further extend to pH-induced complex coacervation in poly-mersomes, we first prepare polymersomes with pH 4 adjusted aqueous core containing 10 mM ATP, 10 mg mL$^{-1}$ PAH, and 0.5 mg mL$^{-1}$ FITC-PAH. Also, 3 μg mL$^{-1}$ of red fluorescent dye (Nile red) is added in the middle oil phase to distinguish the polymersome membrane. Upon increasing the pH from 4 to 7.4 by introducing pH 10 buffered solution near the periphery of polymersomes, we anticipated that ATP within the polymersomes would become negatively charged to com-plementarily interact with the positively charged PAH, resulting in complex coacervation. As expected, changing the pH to 7.4 leads to nucleation of small coacervate droplets followed by maturation via aggregation of these coacervates into a single larger one, as evidenced by the dramatic change from uniform green fluorescence signal inside the polymersomes (Fig. 2b, i) to small coacervate droplets (Fig. 2b, ii) after 45 min. Close examination of this coacervation process reveals that the entire process takes ~100 min after pH adjustment, starting from the sporadic nucleation of coacervate droplets that increases in number but eventually merge into a single large coacervate droplet through 3D-diffusion (Fig. 2c). We note that the matured coacervate droplet dimension depends on the ATP concentration (Fig. 2d) and exhibits an average diameter of $27.4 \pm 3.7$ μm at 10 mM ATP (Fig. 2e and Supplementary Fig. 11). Moreover, we find that the average diameter decreases to $22.5 \pm 3.3$ μm when the pH is increased to pH 9.45 above the pKa value of PAH due to the decrease in the charged amine groups in PAH capable of forming complex coacervates with ATP (Supple-mentary Fig. 12). We note that further increasing the pH condition from 7.4 to pH 11 leads to disassembly of the matured coacervate droplet within the polymersome, as evidenced by the transition from a discrete smaller droplet to larger homogeneous fluorescent signal throughout the polymersome (Supplementary Fig. 13). While these

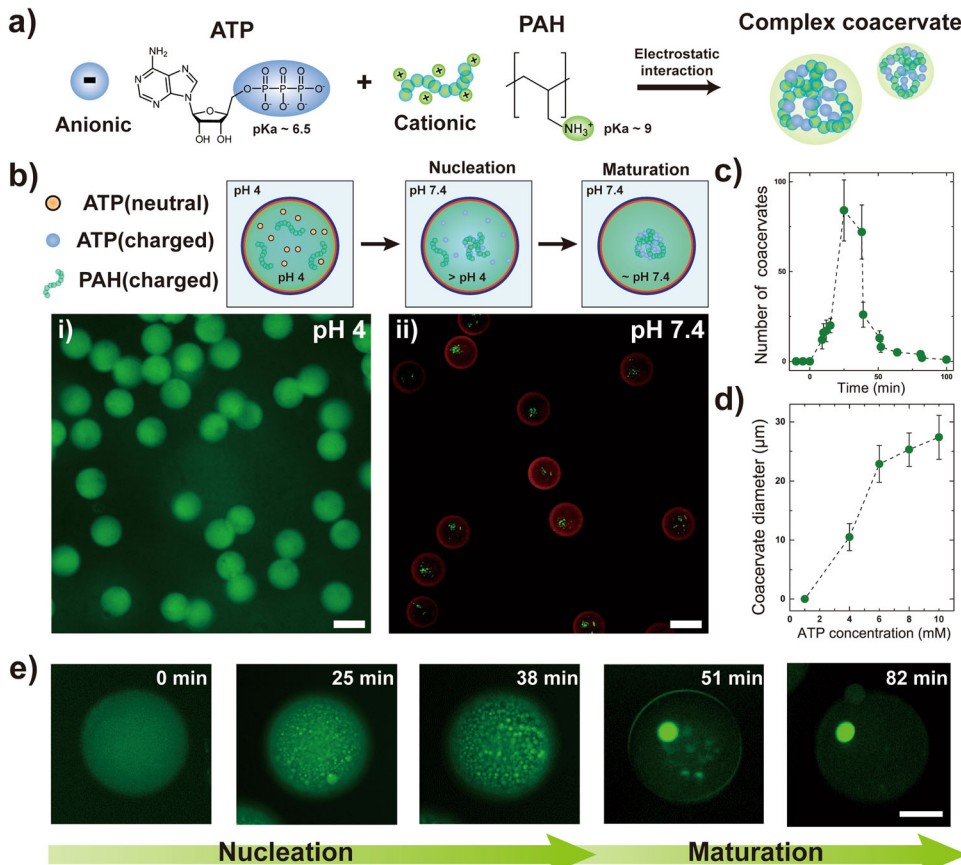

**Fig. 2 | pH-induced complex coacervation in polymersomes. a** Schematics illustrating the complex coacervate formation between adenosine triphosphate (ATP) and poly(allylamine hydrochloride) (PAH). **b** Schematics illustrating the pH-induced complex coacervates in polymersomes. (i) Fluorescence micrograph showing no noticeable formation of complex coacervates within the polymersomes at pH 4. (ii) Fluorescence micrograph showing complex coacervate formation within the polymersomes 45 min after adjusting the pH to 7.4. Both scale bars represent 100 μm. **c** Plot showing the change in the number of coacervate droplets inside polymersomes after increasing the external pH condition to pH 7.4. (n = 11). **d** Plot showing the average diameter of the matured coacervate droplet (t = 120 min) inside the polymersomes with variation in the ATP concentration (n = 26). Data are presented as mean values ± standard deviation, in **c** and **d**. **e** Series of magnified fluorescence micrographs showing the coacervation process from nucleation to maturation. Scale bar represents 100 μm. All error bars represent standard deviation.

results demonstrate that complex coacervates can be induced and disassembled inside Pluronic-based polymersomes by altering the external pH condition, we also noticed that the coacervation process is much slower compared to the proton permeation rate.

To verify the origin of this large discrepancy in time scale of proton diffusion and pH-induced complex coacervation, we investigated the effect of polymersome size and the electrostatic interaction among PAH and ATP. As the coacervation occurs through nucleation of small coacervate droplets followed by maturation into a single larger one, we first studied how the polymersome size affects the time scale of complex coacervation with the assumption that reducing the size would decrease the time scale since the 3D-diffusion length during maturation process becomes shorter. To achieve this, we prepared sets of polymersomes with identical composition but with different sizes and induced complex coacervation by increasing the pH from 4 to 7.4. We observe that the time scale of maturation into a large coacervate droplet decreased by about 20 min and the matured coacervate droplet diameter decreased from 27.4 ± 3.7 to 23.3 ± 2.0 μm when the polymersome size was reduced from 118 to 78.91 ± 4.54 μm (Supplementary Fig. 14).

Next, we studied how the electrostatic interaction between PAH and ATP affects the overall coacervation process. Zeta-potential value of the matured coacervate droplet formed in bulk for 10 mM ATP and 10 mg mL$^{-1}$ PAH was +61.1 ± 3.8 mV, indicating that there exists excess amount of positively charged PAH

compared to ATP with protonated amine groups presented toward the coacervate surface[43]. To investigate whether the presence of excess amount of protonated PAH initially affects the electrostatic interaction with ATP as well as maturation, we induced complex coacervation in the direction of lowering the pH from 10.5 to 7.4, as opposed to the direction of increasing the pH from 4 to 7.4. Unexpectedly, we find that changing the direction of pH transition reduces the time scale of overall complex coacervation process to about 8 to 10 min while the size of the matured coacervate droplet remain similar (27.3 ± 1.58 μm) (Supplementary Fig. 15). The slower coacervation process for the pH transition from 4 to 7.4 case is presumably due to the prevalence of positively charged PAH that impedes the maturation into a single droplet due to repulsive forces, as similarly observed by others for poly-L-lysine and ATP[45]. On the other hand, when the pH is lowered from 10.5 to 7.4, the amine group of PAH gradually protonate to initiate nucleation with ATP, yielding coacervate droplets with decrease in zeta-potential value compared to the case where the pH is increased from 4 to 7.4. Further detailed experimental verifications reveal that the mechanical process of nucleation and maturation is indeed the rate-determining step in the pH-induced complex coacervation (Supplementary Figs. 15–17). While the dimension of the polymersome affects the time scale as well as the resulting matured coacervate droplet size, we also demonstrate that the surface charge of the complex coacervate as well as the direction of pH transition serve as

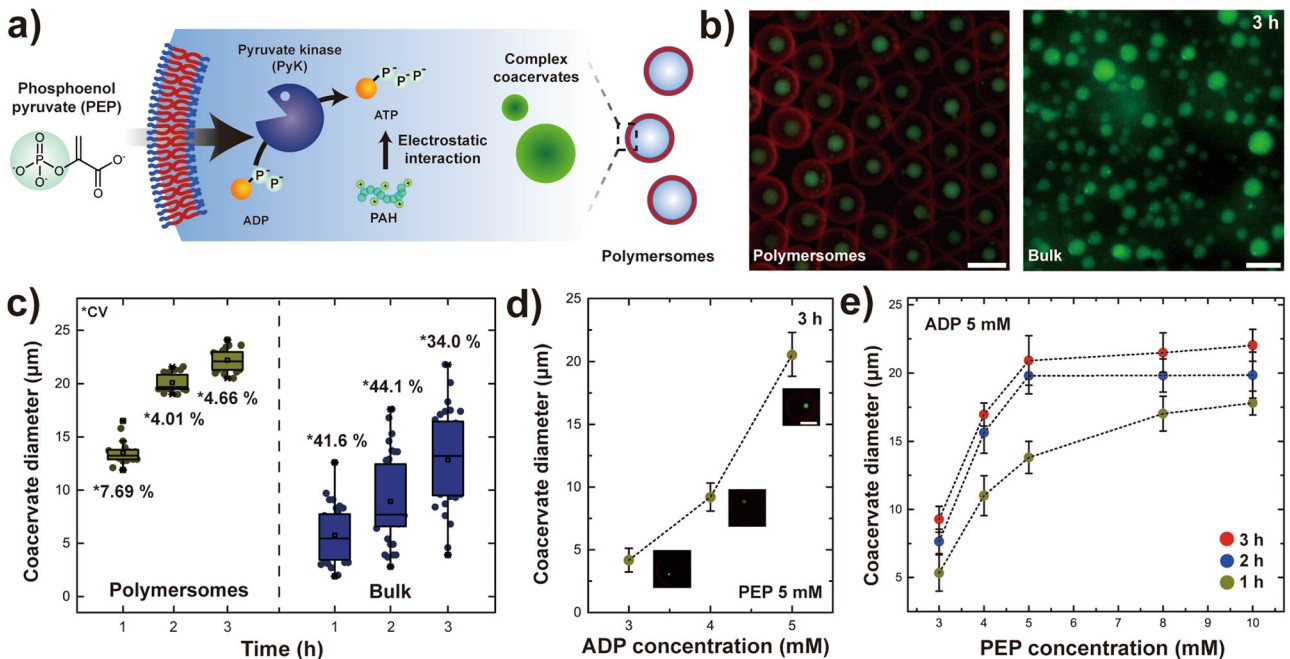

**Fig. 3 | Signal-driven complex coacervation in polymersomes. a** Schematic illustration of the enzymatic reaction-induced complex coacervate formation in polymersomes. **b** Fluorescence micrographs showing the PEP-driven complex coacervate formation via pyruvate kinase (PyK) enzymatic reaction 3 h after PEP infusion in polymersome (left) and in bulk (right). The scale bars each represent 100 μm (left) and 20 μm (right), respectively. **c** Plot showing the change in coacervate diameter over time in polymersomes ($n = 20$) and in bulk ($n = 32$). All box plots indicate median (middle line), 25th, 75th percentile (box) and whiskers as well as outliers (1.5 interquartile range value). **d** Plot showing the coacervate diameter in

polymersomes with variation in ADP concentration ($n = 9$) 3 h after infusion of 5 mM PEP. The inlet fluorescence micrographs show the size of the resulting coacervate within polymersomes at different ADP concentration. Scale bar represent 50 μm. **e** Plot showing the coacervate diameter in polymersomes with variation in PEP concentration at three different time points after infusion of PEP. The red colored circle represent 3 h after PEP infusion while blue and beige each refers to 2 h and 1 h, respectively. Except for a point with 1 h after infusion of 3 mM PEP ($n = 10$), the others have the number of samples as 11 ($n = 11$). All error bars represent standard deviation. Data are presented as mean values ± standard deviation, in **d** and **e**.

a critical factor. We also note that the coacervation process in similar time scale (about 60 min) has been reported by others using pLys and ATP upon changing the pH from 11 to 9[7].

## Signal-driven complex coacervation and cascade enzymatic reaction in polymersomes

While external pH can be used as a cue to trigger complex coacervate formation in Pluronic-based polymersomes, in vitro enzymatic reactions can also be tailored in these polymersomes for signal-driven complex coacervation. By incorporating pyruvate kinase (PyK), an enzyme that catalyzes the last step in glycolysis, a phosphate group from phosphoenolpyruvate (PEP) can be transferred to adenosine diphosphate (ADP), resulting in pyruvate and ATP. As ADP has one less phosphate group than ATP, the electrostatic attraction force with a positively charged polyelectrolyte such as PAH is known to be weak compared to ATP[45–47]. Indeed, performing turbidity assay in bulk for both nucleotides, ADP and ATP, with respect to PAH confirms that ATP with one more phosphate group than ADP can more effectively interact with PAH via electrostatic interactions to form complex coacervates (Supplementary Fig. 18a). In addition, acquisition of the critical salt concentration above which the preformed coacervate comprising of either ATP-PAH or ADP-PAH disassembles (Supplementary Fig. 18b) as well as reversibility experiment (Supplementary Fig. 19) reveals that the electrostatic attraction between PAH and either ADP or ATP can be controlled by the NaCl amount, as they reduce the net interaction between nucleotide and PAH. Based on this study, we chose 5 mM of ADP and 0.2 M NaCl as the optimal condition at which ADP does not form coacervate with PAH while equivalent amount of ATP forms coacervate with PAH. Overall, the PyK catalyzed conversion of ADP to ATP in the presence of PEP allows to tune the electrostatic interaction and thus the complex coacervation behavior. Moreover,

PEP can readily diffuse through the membrane of Pluronic-based polymersome due to low MW (168 Da), serving as a signal to externally drive this enzymatic reaction and yield complex coacervates in polymersomes. We also note that PEP with monophosphate group may also form complex coacervates with PAH but only at high concentration (>17 mM) and appears as a clear single phase at 5 mM (Supplementary Fig. 20). Therefore, we anticipated that usage of PEP as the signal would not interfere with this signal-induced complex coacervation under critical concentration.

To demonstrate this concept of enzymatic reaction-induced complex coacervates in polymersomes detailed in the schematics of Fig. 3a, we first encapsulate pH 7.4 buffer solution containing 5 mM ADP, 10 mg mL⁻¹ PAH, 0.5 mg mL⁻¹ FITC-PAH, and 11.1 unit mL⁻¹ PyK in the polymersomes to induce PEP-driven complex coacervation. We note that PyK is encapsulated with negligible loss and well-retained within the polymersomes (Supplementary Figs. 21 and 22) and that equal concentration of ADP was added in the aqueous continuous phase during polymersome production as well as in the collection bath to prevent the diffusion of ADP (MW 427 Da) across the membrane. 3 μg mL⁻¹ Nile red was also added in the middle oil phase for visualization of the polymersome membrane. After the polymersomes settle in the collection bath, 5 mM PEP is additionally added to result in pyruvate and conversion of ADP to ATP (Fig. 3a). Indeed, we observe successful formation of signal-driven complex coacervates in polymersomes, only after PEP is externally infused into polymersomes (Fig. 3b (left)). This indicates that the PyK within the polymersomes transfers the monophosphate group from PEP to ADP, which leads to production of ATP that can strongly interact with PAH to form ATP-PAH complex coacervates.

To further investigate this PEP-driven enzymatic reaction in confinement provided by the polymersomes, we first compare these

results with the bulk analog. Performing the analogous enzymatic reaction-induced complex coacervation in bulk and monitoring the resulting coacervate droplet dimension change over time in polymersomes and in bulk after PEP infusion reveals that the coacervate droplets formed inside the polymersomes are much uniform and larger in size at all time points compared to the bulk case (Fig. 3b, c). Enzymatically induced ATP-PAH coacervate in polymersomes forms through the similar process as the pH-induced case in which small coacervate droplets initially nucleate and gradually mature into a single large coacervate droplet over time (Supplementary Fig. 23). However, the complex coacervate formation rate is slower compared to the pH-induced analog as this is induced by diffusion of PEP through the membrane followed by enzymatic conversion of ADP to ATP, as evidenced by the time evolution of the coacervate droplet diameter in polymersomes after PEP infusion (Fig. 3c (left)). The fluorescence micrograph of the resulting coacervate droplets in polymersomes after 6 h confirms that enzymatically induced ATP-PAH complex coacervate formation inside polymersomes can be spatiotemporally controlled by external infusion of PEP (Supplementary Fig. 24).

To verify the direct correlation between the reactant concentration on the resulting coacervate dimension in polymersomes, we perform a separate experiment in which we examine the effect of PEP and ADP concentration on the coacervate diameter. When the external PEP concentration is fixed at 5 mM, we find that the diameter of the coacervate within the polymersomes 3 h after PEP infusion increases from $4.2 \pm 0.9$ to $9.2 \pm 1.1$ to $20.9 \pm 1.8 \,\mu m$ as the ADP concentration inside and outside the polymersomes increases from 3, 4, to 5 mM, respectively (Fig. 3d). On the other hand, monitoring the coacervate diameter at three different time points at a fixed ADP concentration of 5 mM as the PEP concentration is systematically varied from 3 to 10 mM shows that the coacervate diameter does not increase above 5 mM PEP concentration after maturation and that the initial amount of PEP present outside the polymersomes governs the rate of coacervate formation (Fig. 3e). Overall, these results indicate that the Pluronic-based membrane in polymersomes serve as a diffusion barrier for the PEP and that stoichiometric ratio of the PEP and ADP can be tuned to precisely control the reaction kinetics as well as the dimension of the coacervate formed within the polymersomes.

## Signal-driven cascade reaction in polymersomes

To exploit the PEP-driven enzymatic reaction in polymersomes for reconstituting the highly combinatorial and interconnective enzymatic reactions in cells, we prepare polymersomes containing 15.0 unit mL$^{-1}$ PyK, 30.0 unit mL$^{-1}$ lactic dehydrogenase (LDH), and 10 mM of ADP and nicotinamide adenine dinucleotide (NADH). LDH (MW 140 kDa) and NADH (MW 663 Da) are incorporated within the polymersomes to induce cascade enzymatic reaction upon external infusion of PEP (Fig. 4a). LDH catalyzes the conversion of NADH into NAD$^+$ in the presence of pyruvate, which is the product from the PyK enzymatic reaction. As a result, when PEP permeates through the polymersome membrane, the PyK catalyzed conversion of ADP to ATP simultaneously generates pyruvate which now act as the reactant to yield lactate and converts NADH into NAD$^+$ during this process. Due to autofluorescence of NADH ($\lambda_{em} = 460$ nm) and the possibility of photobleaching over time, the difference in fluorescence intensity reduction with and without PEP infusion can be regarded as the conversion of NADH to NAD$^+$ by the cascade enzymatic reaction (Fig. 4b). Indeed, we find that the blue fluorescence signal from NADH more rapidly decays upon infusion of PEP adjacent to polymersomes compared to the non-infused analog, indicating the successful implementation of enzymatic cascade reactions in our Pluronic-based polymersomes.

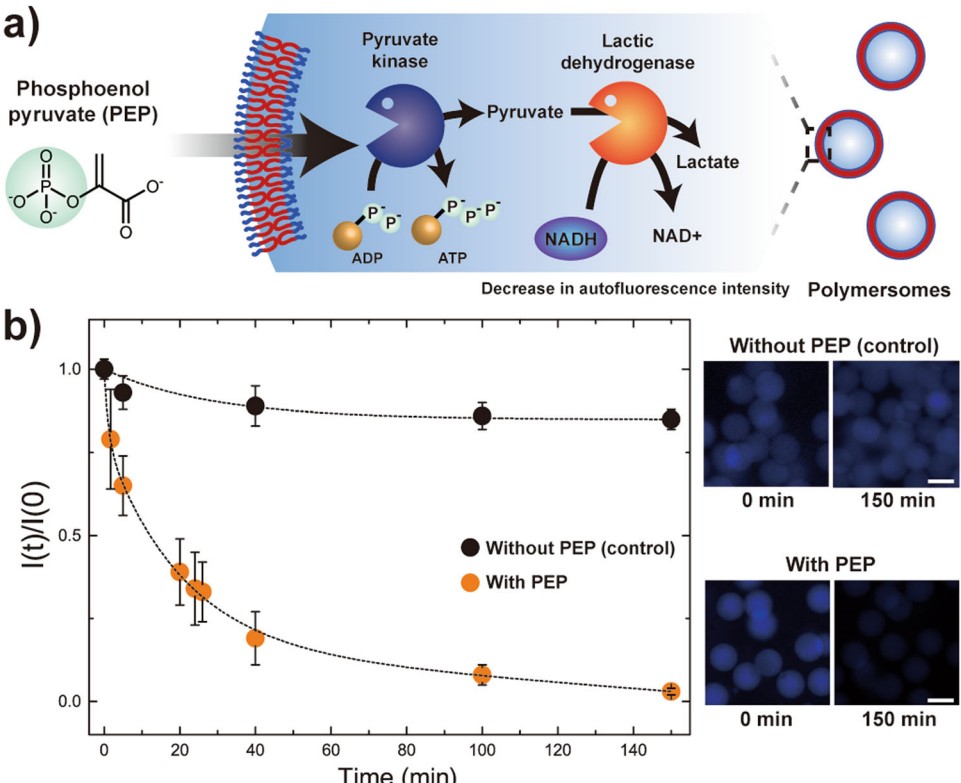

**Fig. 4 | Signal-driven cascade enzymatic reaction in polymersomes. a** Schematic illustration of the PEP-driven cascade enzymatic reaction in polymersomes by incorporation of both pyruvate kinase (PyK) and lactic dehydrogenase (LDH). **b** Plot and fluorescence micrographs comparing the decrease in fluorescence intensity with and without infusion of PEP ($n = 16$). Scale bars in the fluorescence micrographs represent 100 μm. All error bars represent standard deviation.

### Signal-driven actin polymerization in polymersomes

To demonstrate the utility of PEP-driven enzymatic reaction in polymersomes as a responsive synthetic cellular model, we further extend this concept to signal-driven reconstitution of cytoskeleton. To achieve this, we use globular actin (G-actin) proteins for building cell cortex inside the polymersomes via signal-driven enzymatic reaction. G-actin can be polymerized in an in vitro physiological ion concentration to become filamentous actin (F-actin), a helical actin fiber, due to the different affinity of actin to ATP and ADP. The conversion of G- to F-actin is accompanied by conformational change due to hydrolysis of ATP and ATP-actin exhibit stronger affinity than ADP-actin, resulting in the ADP to easily fall apart from actin monomer. As a result, one end of the polymerized F-actin easily dissociates ADP-actin monomer, and the disassembled ADP-actin can be converted back to ATP-actin via PyK enzymatic reaction which can easily bind to the other end of the polymerized F-actin[48]. Therefore, introducing PEP in the exterior of the polymersomes induces the PyK enzymatic reaction to facilitate the elongation of actin filament within the polymersomes (Fig. 5a). However, actin monomers can also initiate polymerization in the presence of $K^+$ and $Mg^{2+}$ by counter-ion condensation[49–51]; the critical concentration (CC) value above which the actin monomer starts to spontaneously polymerize decreases by two-orders of magnitude in the presence of 2 mM $Mg^{2+}$ and 50 mM $K^+$ (CC = 0.03 mg mL$^{-1}$) compared to ion-free condition (CC = 3.0 mg mL$^{-1}$)[52]. Thus, we initially do not include $K^+$ and $Mg^{2+}$ in the aqueous core and instead introduce them with the PEP to control the onset of actin polymerization.

To achieve PEP-driven actin polymerization in polymersomes for structural reconstitution of cytoskeleton, we encapsulate an aqueous buffer solution without $Mg^{2+}$ and $K^+$ but contains 10 mM ADP, 0.1 mg mL$^{-1}$ G-actin, 11.1 unit mL$^{-1}$ PyK, and 0.1 μM green fluorescent dye labeled phalloidin (Alexa Flour 488-conjugated phalloidin) which selectively stains the F-actin in polymersomes. As phalloidin exhibit high Pearson's correlation coefficient for the linear actin structure, fluorescence signal from phalloidin can be regarded as successful polymerization of G-actin into F-actin[53]. Then, we introduce an aqueous buffer solution containing 20 mM PEP, 150 mM $K^+$, and 50 mM $Mg^{2+}$ at the exterior of polymersomes to initiate the ATP-dependent actin polymerization. While no apparent nucleation event is initially observed in the pristine polymersomes ($t = 0$ min), actin nucleation and elongation occur only after the buffer solution is injected as the signal (Fig. 5b). Upon addition, successive influx of PEP, $K^+$, and $Mg^{2+}$ initiate G-actin nucleation and concatenated polymerization with other G-actin monomers to exhibit increase in fluorescence intensity resulting from phalloidin. The fluorescence intensity rapidly increases due to continuous polymerization of G-actin to F-actin, which eventually plateaus after about 90 min.

To verify whether this intensity increase is due to PEP-driven enzymatic reaction followed by actin polymerization, we prepare two sets of polymersomes without PEP, $K^+$, and $Mg^{2+}$, but contains PyK, G-actin, and phalloidin in the aqueous core. Then, we externally infuse a buffer solution that contains $K^+$ and $Mg^{2+}$ and PEP adjacent to polymersomes (referred as PEP) in one set while identical buffer solution without the PEP (referred as control) is infused in the other. We observe significant increase in fluorescence intensity and clear filament structure formation within the polymersome for the set with all three signals. On the contrary, no discernible filament structure and fluorescence intensity was observed in the control that contains $K^+$ and $Mg^{2+}$ but is without PEP. These results clearly demonstrate that the PEP-driven enzymatic conversion of ADP into ATP inside the polymersomes increases the ATP-actin concentration, which leads to nucleation and elongation of actin filaments (Fig. 5b, c).

To assess the role of each signal in PEP-driven actin polymerization within the polymersomes, we monitor the formation of F-actin structure in four sets of polymersomes with different signal conditions

(Fig. 5d). Indeed, in the case where either $Mg^{2+}$ or $K^+$ are absent, we observe no distinct actin polymerization in the polymersomes, indicating that ATP cannot be produced by PyK in the absence of metal ion cofactors or reach the critical concentration for G-actin nucleation. In addition, for the condition at which $Mg^{2+}$ and $K^+$ were both present but without PEP, we also do not observe any increase in fluorescence intensity or any evidence of F-actin formation. On the other hand, when all three signals are simultaneously infused, we observe that the PEP-driven enzymatic reaction become active, as evidenced by the increase in fluorescence intensity and the successful formation of F-actin structure within the polymersomes (Fig. 5d and Supplementary Figs. 25 and 26). Overall, our results reveal the interplay between in vitro signal-driven enzymatic reactions and reconstitution of actin filaments, enabled by the semi-permeable nature of the Pluronic-based polymersome membrane.

## Discussion

Herein, we present a microfluidic strategy to spatiotemporally control enzymatic reaction in artificial cell-like polymersomes without the usage of biopores. These Pluronic-based polymersomes are stable and semi-permeable, allowing controlled transport of protons and small molecules across the polymersome membrane for pH-induced assembly of complex coacervates. In addition, we utilize the PEP-driven PyK enzymatic reaction for the conversion of ADP to ATP, which alters the electrostatic attraction between PAH to selectively form ATP-PAH complex coacervates. Moreover, we extend this signal-driven enzyme reaction for cascade reaction and reconstituting cytoskeleton in polymersomes by polymerizing actin monomers into filament networks.

As the strategy outline in this work is applicable to broad range of amphiphilic block copolymers due to advances in controlled polymer synthesis, the membrane composition can be fine-tuned to control the permeability without the use of biopores. Indeed, preparing analogous sets of polymersomes with different compositions (Pluronic L121 and L61 blended polymersomes and poly(butadiene)-b-poly(ethylene oxide) (PB-PEO) polymersomes) and comparing the fluorescence intensity change with Pluronic L121-based polymersomes reveal that the membrane permeability can be either increased or decreased by mixing with lower molecular weight analogs or utilizing high f value polymer. (See the details in Supplementary information Fig. S27 and Notes). However, we note that in case where exchange of larger biomolecules with molar masses above 1 kDa is needed between two different compartments for the spatiotemporal control of biochemical reactions, membrane proteins can be incorporated into the membrane to transport these biomolecules, as reported previously by others[16,21,51,54,55]. Furthermore, since the model reactions demonstrated in this work can be further extended to other enzymatic reactions and external signals to impart functionality, we expect that the spatiotemporal control offered by the Pluronic-based polymersomes provides an opportunity in the bottom-up approach to design responsive and programmable artificial cell-like systems that were previously difficult to achieve.

## Methods

### Materials

Pluronic L121, Pluronic L61, glycerol, sucrose, KCl, MgCl$_2$, CaCl$_2$, 4-(2-hydroxyethyl)piperazine-1-ethanesulfonic acid (HEPES > 99.5%), 8-hydroxypyrene-1,3,6-trisulfonic acid trisodium salt (HPTS > 96%), carboxyfluorescein, sulforhodamine B, rhodamine 6 G, eriochrome black T, polyethylene glycol (PEG, MW 6000 Da), poly(vinyl alcohol) (PVA, MW 13000–23000 Da, 87–89% hydrolyzed), poly(allylamine hydrochloride) (MW 17500 Da), FITC, NHS-Fluorescein, ATP disodium salt, ADP sodium salt, valinomycin (>98%), pyruvate kinase from rabbit muscle (Type III, 350–600 units mg$^{-1}$ protein), lactic dehydrogenase from rabbit muscle (Type XI, 600-1,200 units mg$^{-1}$ protein), Bovine

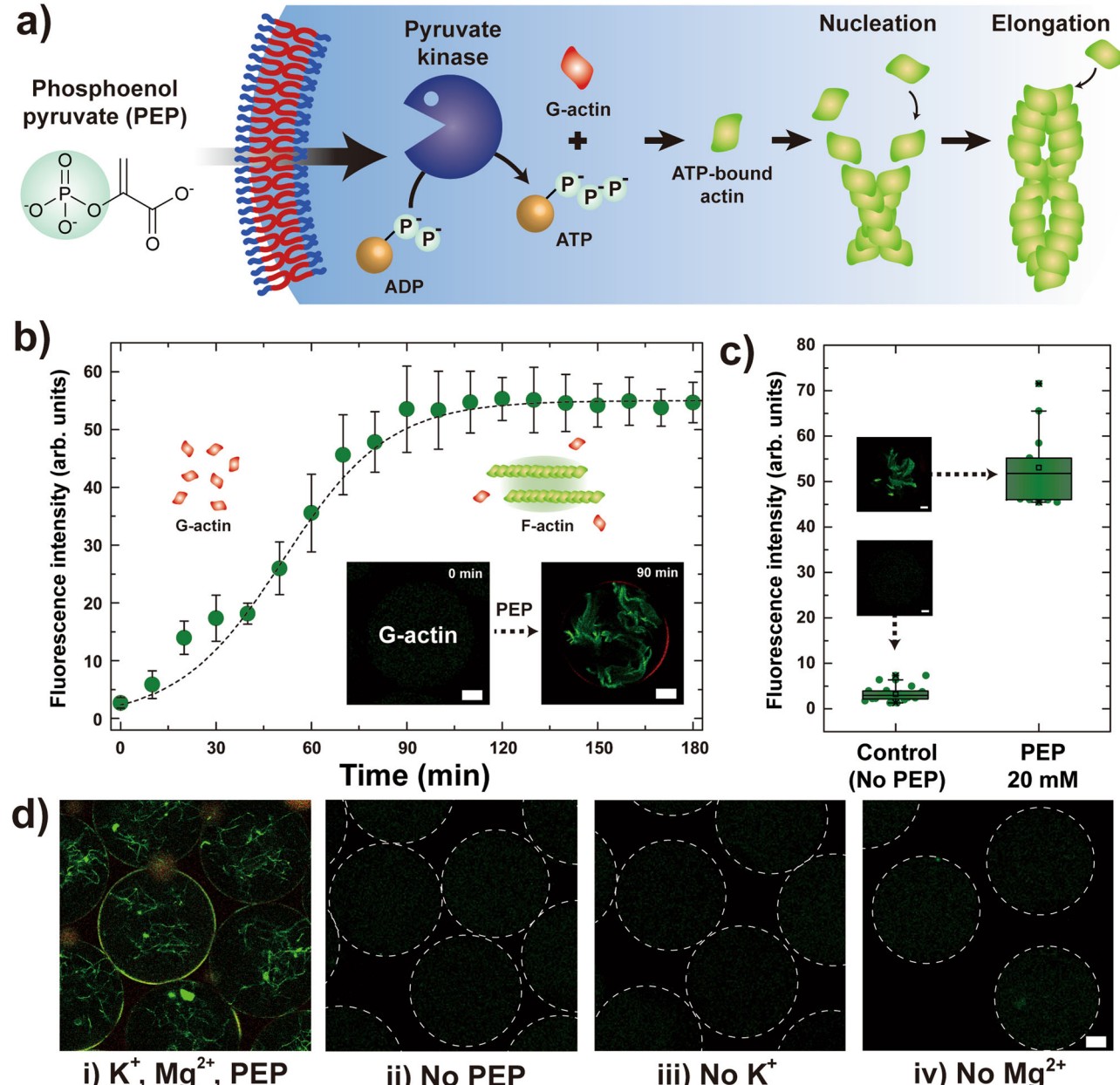

**Fig. 5 | Signal-driven actin polymerization in polymersomes. a** Schematics illustrating the PEP-driven actin polymerization in polymersomes. Globular actin (G-actin) is polymerized to become filamentous actin (F-actin) in the presence of ATP. **b** Plot and schematic illustration showing the fluorescence intensity change during the actin nucleation and elongation after PEP is infused as the signal. The F-actin and polymersome membrane are each stained with Alexa Fluor 488-phalloidin (green) and Nile Red (red), respectively ($n = 14$). The inset fluorescence micrographs show actin elongation within the polymersome 90 min after PEP infusion. Scale bar represent 20 μm. Data are presented as mean values ± standard deviation. **c** Plot showing the fluorescence intensity of polymersomes 120 min after PEP infusion ($n = 14$) compared to the control without PEP infusion ($n = 28$). Scale bars of the inset micrographs are 10 μm. Box plots indicate median (middle line), 25th, 75th percentile (box) and whiskers as well as outliers (1.5 interquartile range value). **d** Sets of confocal micrographs showing the interior of the polymersomes at different signal conditions. (i) 20 mM PEP, 150 mM $K^+$, 50 mM $Mg^{2+}$; (ii) No PEP, 150 mM $K^+$, 50 mM $Mg^{2+}$; (iii) 20 mM PEP, No $K^+$, 50 mM $Mg^{2+}$; (iv) 20 mM PEP, 150 mM $K^+$, no $Mg^{2+}$. Scale bars represent 20 μm and the membrane of polymersomes are highlighted in white dotted circles for visualization purposes. All error bars represent standard deviation.

Serum Albumin (BSA, Product No. A7030), Bradford Reagent (Product No. B6916), NADH (reduced disodium salt, Grade II), DL-dithiothreitol (DTT > 98.0% for molecular biology), chloroform, cyclohexane, n-octadecyltrimethoxysilane, HCl (36.5–38.0%), NaOH (>98.0%, ACS reagent) were purchased from Sigma Aldrich. 2-[methoxy(polyethyleneoxy) propyl]trimethoxy silane was purchased from Gelest. Phosphoenolpyruvate was purchased from Roche. Actin (>99%, rabbit muscle) was purchased from Cytoskeleton. Pluronic F-68 (Poloxamer P188), Atto 633 NHS ester, Atto 590, and Alexa Fluor 488 conjugated phalloidin were

purchased from ThermoFisher. Poly(butadiene)-b-poly(ethylene glycol) (PB-PEO) was purchased from PolymerScience. (Canada).

**Solution compositions**

The inner aqueous core of the polymersomes exhibit the following solution composition for the experiments described in each figure: 15 vol% glycerol, 16.7 mM PEG, 150 mM KCl, 25 mM HEPES (pH 7.4), 5 mM $MgCl_2$. (Fig. 1a, and Supplementary Fig. 2); 10 wt% PEG (Supplementary Fig. S3); 1.43 ± 0.11 mg mL$^{-1}$ FITC-PyK stock solution was diluted in

20.0 mM PEG, 180 mM KCl, 30 mM HEPES (pH 7.4), 6 mM MgCl$_2$ by a volume ratio of 1:5, leading to final concentrations of 0.24 mg mL$^{-1}$ FITC-PyK, 16.7 mM PEG, 150 mM KCl, 25 mM HEPES (pH 7.4), 5 mM MgCl$_2$. (Supplementary Fig. 4 and 21); 10 wt% PEG, and 5 µg mL$^{-1}$ of each fluorescent dyes (Figs. 1d, 1e, and Supplementary Fig. 6, and 27); 10 wt% PEG (Fig. 1f); 10 wt% PEG, 1 mM eriochrome black T, adjusted to pH 10 using 6 M NaOH (Supplementary Fig. 7b); 15 vol% glycerol, 16.7 mM PEG, 150 mM KCl, 25 mM HEPES, 5 mM MgCl$_2$, and 5 µg mL$^{-1}$ HPTS, adjusted to pH 4 using 1 M HCl (Supplementary Fig. 8); 15 vol% glycerol, 16.7 mM PEG, 150 mM KCl, 25 mM HEPES, 5 mM MgCl$_2$, 10 mg mL$^{-1}$ PAH, 0.5 mg mL$^{-1}$ FITC-PAH, adjusted to pH 4 using 1 M HCl (Supplementary Fig. 10); 15 vol% glycerol, 16.7 mM PEG, 150 mM KCl, 25 mM HEPES, 5 mM MgCl$_2$, 10 mM ATP, 10 mg mL$^{-1}$ PAH, 0.5 mg mL$^{-1}$ FITC-PAH, adjusted to pH 4 using 1 M HCl (Figs. 2b-2e, Supplementary Fig. 11-13, and 14); 15 vol% glycerol, 16.7 mM PEG, 150 mM KCl, 25 mM HEPES, 5 mM MgCl$_2$, 10 mM ATP, 10 mg mL$^{-1}$ PAH, 0.5 mg mL$^{-1}$ FITC-PAH, adjusted to pH 10.5 using 6 M NaOH. (Supplementary Fig. 15); 15 vol% glycerol, 16.7 mM PEG, 150 mM KCl, 25 mM HEPES, 5 mM MgCl$_2$, 10 mM ATP, 5 mg mL$^{-1}$ PAH, 0.25 mg mL$^{-1}$ FITC-PAH, adjusted to pH 4 using 1 M HCl. (Supplementary Fig. 16); 15 vol% glycerol, 16.7 mM PEG, 150 mM KCl, 25 mM HEPES, 5 mM MgCl$_2$, 5 mM ATP, 10 mg mL$^{-1}$ PAH, 0.5 mg mL$^{-1}$ FITC-PAH, adjusted to pH 10.5 using 6 M NaOH. (Supplementary Fig. 17); 15 vol% glycerol, 16.7 mM PEG, 150 mM KCl, 0.2 M NaCl, 25 mM HEPES (pH 7.4), 5 mM MgCl$_2$, 5 mM ADP, 10 mg mL$^{-1}$ PAH, 0.5 mg mL$^{-1}$ FITC-PAH, 11.1 unit mL$^{-1}$ pyruvate kinase (Fig. 3b, c, e, Supplementary Fig. 23 and 24); 15 vol% glycerol, 16.7 mM PEG, 150 mM KCl, 0.2 M NaCl, 25 mM HEPES (pH 7.4), 5 mM MgCl$_2$, varying concentration of ADP (3, 4, 5 mM), 10 mg mL$^{-1}$ PAH, 0.5 mg mL$^{-1}$ FITC-PAH, 11.1 unit mL$^{-1}$ pyruvate kinase (Fig. 3d); 15 vol% glycerol, 16.7 mM PEG, 150 mM KCl, 0.2 M NaCl, 25 mM HEPES (pH 7.4), 5 mM MgCl$_2$, 10 mM ADP, 10 mM NADH, 15.0 unit mL$^{-1}$ pyruvate kinase, 30.0 unit mL$^{-1}$ lactic dehydrogenase (Fig. 4b); 15 vol% glycerol, 16.7 mM PEG, 0.2 mM CaCl$_2$, 20 mM Tris-HCl (pH 7.4), 10 mM ADP, 11.1 unit mL$^{-1}$ pyruvate kinase, 0.1 mg mL$^{-1}$ actin monomer, 0.1 µM Alexa Fluor 488-phalloidin, 2 mM DTT (Fig. 5b–d, Supplementary Fig. 25 and 26). To prevent fungal contamination, we autoclave the samples for 20 min at 120 °C and 1 atm or purify through a vacuum filter with 0.2 µm (Corning®) prior to dissolving proteins, enzymes, fluorescent dye molecules, and dye-tagged molecules. The stock solutions for actin polymerization were used immediately after preparation.

For the experiments described in each figure, the aqueous continuous phase contains the followings: 15 vol% glycerol, 90 mg mL$^{-1}$ PVA, 150 mM KCl, 25 mM HEPES (pH 7.4), 5 mM MgCl$_2$. (Fig. 1a, and Supplementary Fig. 2); 10 wt% PVA, 0.5 wt% Poloxamer P188 (Supplementary Fig. 3); 100 mg mL$^{-1}$ PVA, 150 mM KCl, 25 mM HEPES (pH 7.4), 5 mM MgCl$_2$. (Supplementary Fig. 4, and 21); 10 wt% PVA (Fig. 1d–f, Supplementary Fig. 6, 7b and 27); 15 vol% glycerol, 90 mg mL$^{-1}$ PVA, 150 mM KCl, 25 mM HEPES, 5 mM MgCl$_2$, adjusted to pH 4 using 1 M HCl. (Supplementary Fig. 10); 15 vol% glycerol, 90 mg mL$^{-1}$ PVA, 150 mM KCl, 25 mM HEPES, 5 mM MgCl$_2$, adjusted to pH 4 using 1 M HCl. (Figs. 2b-2e, Supplementary Fig. 11-14 and 16); 15 vol% glycerol, 90 mg mL$^{-1}$ PVA, 150 mM KCl, 25 mM HEPES, 5 mM MgCl$_2$, adjusted to pH 10.5 using 6 M NaOH. (Supplementary Fig. 15 and Fig. 17); 15 vol% glycerol, 90 mg mL$^{-1}$ PVA, 150 mM KCl, 0.2 M NaCl, 25 mM HEPES (pH 7.4), 5 mM MgCl$_2$, 5 mM ADP (Figs. 3b, c, 4e, Supplementary Fig. 23 and 24); 15 vol% glycerol, 90 mg mL$^{-1}$ PVA, 150 mM KCl, 0.2 M NaCl, 25 mM HEPES (pH 7.4), 5 mM MgCl$_2$, and three different ADP concentration as in the main text. (Fig. 3d); 15 vol% glycerol, 90 mg mL$^{-1}$ PVA, 150 mM KCl, 25 mM HEPES (pH 7.4), 5 mM MgCl$_2$, 10 mM ADP (Fig. 4b); 15 vol% glycerol, 90 mg mL$^{-1}$ PVA, 0.2 mM CaCl$_2$, 20 mM Tris-HCl (pH 7.4), 10 mM ADP, 2 mM DTT (Fig. 5b–d, Supplementary Fig. 25 and 26). For the experiments that require pH change, the aqueous continuous phase was adjusted to the same condition as the inner aqueous phase using 1 M HCl or 6 M NaOH. For actin polymerization, we used the following stock solution

for the signal: 15 vol% glycerol, 16.7 mM PEG, 0.2 mM CaCl$_2$, 20 mM Tris-HCl (pH 7.4), 10 mM ADP, 300 mM KCl, 100 mM MgCl$_2$, 40 mM PEP, 2 mM DTT (Fig. 5b–d, Supplementary Figs. 25 and 26); depending on the signal conditions described in the figure, relevant substances are excluded from this solution composition.

The collection bath for the polymersomes consists of 15 vol% glycerol, 150 mM KCl, 25 mM HEPES (pH 7.4), 5 mM MgCl$_2$. Also, for the condition without Mg$^{2+}$ and K$^+$, 15 vol% glycerol, 200 mM NaCl, 25 mM HEPES (pH 7.4) is used instead. Any osmotic imbalance was adjusted using glucose, and pH was further adjusted using 1 M HCl or 6 M NaOH. Additional substances (e.g. ADP, ATP, or salts) related to the enzymatic reaction are described in the main text.

## Fabrication of microfluidic device

We use two types of glass capillaries, a square capillary with an inner width of 1.05 mm and a cylindrical capillary with an outer diameter of 1.00 mm (Atlantic International Technology, Inc.). To fabricate the microfluidic device, a square capillary with a length of 30 mm was first attached to a glass microscope slide (LK Lab Korea, 76 × 52 mm) using 5 min Epoxy (Devcon). Then, two cylindrical capillaries that were tapered using a micropipette puller (P-97, Sutter Instrument) were further polished with a sandpaper to each have an outer diameter of 130 and 200 µm, respectively. The cylindrical capillary with the larger diameter (200 µm) is used as the collection capillary and was modified with 2-[methoxy(polyethyleneoxy)propyl]-trimethoxy silane (Gelest, Inc.) to make both the inner and the outer surface hydrophilic. For the other cylindrical capillary, defined as the injection capillary (130 µm), the inner surface of the injection capillary was treated with n-octadecyltrimethoxysilane (Sigma Aldrich) to make the surface hydrophobic while the outer surface of the injection capillary was made hydrophilic. Another cylindrical capillary whose outer diameter is smaller than the inner diameter of the injection capillary was prepared using a burner and polished to have an outer diameter of 100 µm. After attaching the square capillary on to the microscope glass, the injection capillary and the collection capillary were inserted from the opposite ends into the square capillary and fixed using 5 min Epoxy. After appropriate alignment of these capillaries, we then insert the smallest capillary (100 µm) into the injection capillary, followed by attachment of the dispensing needles on top for injection of each fluids comprising the emulsion droplet (Supplementary Fig. 1).

## Image acquisition and operation of the microfluidic device for stable production of polymersomes

To make the double emulsion droplet templates for polymersomes, we use syringe pumps (KDS Legato 100) to control the fluid flow rates. The flow rates for the inner aqueous phase (W$_1$), middle oil phase (O), and the aqueous continuous phase (W$_2$) were set at the values of 500, 500, and 8000 µl h$^{-1}$, respectively. For the collection bath, we prepared a large glass container (70 mm × 50 mm × 20 mm) using glass microscope slides (LK Lab Korea, 76 × 52 mm) to facilitate the evaporation of chloroform during dewetting transition. The double emulsion droplet templates produced from the device are directly collected in this container filled with excess amount of aqueous continuous phase. All experiments were performed at room temperature. The polymersomes produced were monitored using an inverted microscope (Eclipse Ts2, Nikon) equipped with a high-speed camera (FastCam Mini UX50, Photron). Fluorescence micrographs of the polymersomes were acquired using an inverted microscope (Eclipse Ti2, Nikon) equipped with a CMOS camera (Zyla 5.5, Andor).

## Synthesis of Fluorescein Isothiocyanate (FITC) labeled PAH

We synthesized FITC labeled PAH using the previously reported method[56,57]. Briefly, we dissolve 20 mg of FITC in 2.5 mL of dimethyl sulfoxide (DMSO). In a separate vial, 2.5 g of PAH (M.W. ~17500 Da) was dissolved in 30 mL of 18.2 MΩ·cm deionized (DI) water and adjusted to

pH of 9 using 6 M NaOH. Then, FITC in DMSO solution is carefully poured into PAH solution and stirred for 2 days in the dark at room temperature. The resulting solution was subsequently dialyzed to DI water with 7 kDa MW cut-off tubing (SnakeSkin, ThermoFisher) for 7 days. After dialysis, solution was filtered with 0.2 μm PTFE syringe filter (ThermoFisher) prior to usage.

## Turbidity assay

Turbidity (%) was calculated by measuring the absorbance at the wavelength of 600 nm using a microplate reader (Wallac 1420 Victor2 Microplate Reader version 3.00 Revision 5, PerkinElmer) after mixing the two solutions each containing PAH and nucleotides. Briefly, all mixture samples contain 15 vol% glycerol, 25 mM polyethylene glycol (MW 6000 Da), 150 mM KCl, 25 mM HEPES, 5 mM $MgCl_2$, 10 mg mL$^{-1}$ PAH (MW 17500 Da), and variation in the ADP or ATP concentration (0–20 mM) after which they are adjusted to pH of 7.4 using 6 M NaOH. Each well in the 96-microplate contains 100 μL of the samples, and the samples are read 10 times. Absorbance was measured after shaking the plate for 0.3 s at 24 ± 1 °C and the turbidity was calculated using the relation that turbidity (%) is $100(1\text{-}10^{-\text{Absorbance}})$. For the acquisition of the critical salt concentration above which the coacervate disassembles, we measure the absorbance for each sample while varying the salt concentration using 4 M NaCl solution and determine the point at which the absorbance becomes zero. Turbidity plot for PEP and PAH is obtained in the same manner.

## Optical density change in complexometric titration

Observation of overall color change inside polymersomes after injection of $Mg^{2+}$ solution was captured with a stereoscopic zoom microscope (SMZ800N, Nikon and TCapture version 4.3.0). Complexometric titration using Eriochrome black T was measured using a Hidex Sense Microplate Reader (Hidex Software 425-301, Finland). Freshly prepared polymersomes were collected and allowed to sink stably in a 96-well plate. After suction of the remaining supernatant on the submerged polymersomes, the $Mg^{2+}$ aqueous solution was gently injected, and measured using Hidex after 10 min. Measurements were made at 23 °C, and the control group and $Mg^{2+}$ addition group were measured in 12 wells each.

## Characterization of actin polymerization in polymersomes

Actin polymerization was observed using a confocal microscope (STELLARIS 5 Confocal Microscope, and LAS X software version 4.1.0, Leica). Signal conditions at which the structure of actin filaments forms or not were all assessed using the confocal microscope. Nile red-stained polymersome membrane was observed with an Alexa Fluor 633 filter, and selective staining of F-actin by phalloidin was observed with Alexa Fluor 488 filter.

## Determination of fluorescent dye molecule and ion permeability of Pluronic-based membrane in polymersomes

The permeability of each fluorescent dye molecule was estimated by adapting the simplified solution of Fick's law and fitting the normalized fluorescence intensity change (I(t)/I(0)) data from Fig. 1d, e, which is related to the normalized concentration change of the diffusing species, C(t)/C(0)[28,31]. The permeability of each fluorescent dye molecule (P) can be calculated using the following relation:

$$\frac{I(t)}{I(0)} = \frac{C(t)}{C(0)} = \exp\left(-\frac{3*P}{R}t\right) \qquad (1)$$

where the R is the average radius of polymersome. To determine the ion permeability ($P_{ion}$) of polymersome, we adopted the method described by other[28], in which volumetric swelling of polymersome is monitored under osmotic pressure in the presence of permeable ions

($K^+$, $Mg^{2+}$). The osmolyte concentrations of 10 wt% PEG solution within the aqueous core, 3.72 wt% KCl solution, 3.12 wt% $MgCl_2$ solution are each 100 ± 3 mOsm kg$^{-1}$, 888 ± 5 mOsm kg$^{-1}$, 897 ± 3 mOsm kg$^{-1}$, respectively. The change in normalized vesicle radius with time acquired for both hypertonic KCl and $MgCl_2$ solutions were fitted with the linearized vesicle swelling formula shown below:

$$\left(\frac{R}{R_0}\right)^4 = 1 + \frac{4*P_{ions}}{R_0}t \qquad (2)$$

where R is vesicle radius at variant time, $R_0$ is initial vesicle radius (55 ± 3 μm), and $P_{ions}$ is the ion permeability.

## Statistics and reproducibility

All experiments were performed at least as technical triplicates. All quantitative data are presented as average values with error bars which represent ± standard deviation (SD). The average values, SD, and slope were assessed using Origin 8.0 software.

## Reporting summary

Further information on research design is available in the Nature Research Reporting Summary linked to this article.

## Data availability

The generated data supporting the findings of this study are provided in the paper, Supplementary Information an uploaded on Figshare (https://doi.org/10.6084/m9.figshare.20462466). Data is also available from the corresponding author upon request. Source data are provided with this paper.

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

## Acknowledgements

This work was supported by the National Research Foundation of Korea (NRF) grant funded by the Korea government (MSIT) (No. 2020R1C1C1004642), (No. 2019K1A4A7A02113715), (No. 2021R1A4A1021972), and the grant of the Korea Health Technology R&D Project through the Korea Health Industry Development Institute (KHIDI), funded by the Ministry of Health & Welfare, Republic of Korea (No. HP20C0006).

## Author contributions

H.S. and H.L. conceived the project, designed the experiments. H.S. performed the experiments and the data analysis. H.S. and H.L. wrote the paper.

## Competing interests

The authors declare no competing interests.
