## [Peer Review File · Nature Communications]

Spatiotemporal Control of Signal-Driven Enzymatic Reaction in Artificial Cell-Like PolymersomesREVIEWER COMMENTS

Reviewer #1 (Remarks to the Author):

This manuscript reports artificial cell-like polymersomes for controlled enzymatic reactions. The pluronic-based polymersomes were designed by microfluidics, of which membrane has a molecular cut-off threshold of permeation around 500 Da and allows the diffusion of metal ions. The authors encapsulated pyruvate kinase as well as ADP and PAH in the polymersomes. When the polymersomes were suspended in the solution of phosphoenol pyruvate (PEP), phosphoenol pyruvate diffuses through the membrane and the pyruvate kinase catalyzes the transfer of a phosphate group from PEP to ADP to yield ATP. As PAH and ATP produce a complex coacervate through electrostatic attraction, an inner compartment is formed inside the polymersome. Similarly, filamentous actin is produced through ATP-actin binding and polymerization when the polymersome contains globular actin. With encapsulation of pyruvate kinase and lactic dehydrogenase as well as ADP and NADH, a PEP-driven cascade enzymatic reaction is demonstrated. I read this manuscript with great interest and I believe that the polymersome in this work is promising as a platform for artificial cells. Therefore, I recommend the publication of this work in Nature Communications and hope the following minor comments help the authors improve their work:

1. In Figures 1d and e, is there no change in the size of the polymersome? The permeability might be overestimated if there was an inward flow of water due to imperfect osmolarity matching between the lumen and surrounding (hypotonic).
2. In Figure 1f, it is suggested to discuss why the permeabilities of K^+ and Mg^{2+} ions are much smaller than those of small dyes. Are the hydrodynamic diameters of K^+ and Mg^{2+} ions larger due to hydration shells? Or is there any other molecular interaction to reject the ions?
3. In Figure 2b, is it necessary to increase pH up to 10 for the conservation? From pKa values of ATP and PAH, it seems that pH near 7-8 is appropriate to induce electrostatic attraction. Is there any influence of pH on the coacervate diameter and the time scale for coacervation? I also wonder whether the polymersomes remain stable for the range of pH from 4 – 10.
4. It is suggested to increase the color contrast for symbols in Figure 5b.

Reviewer #2 (Remarks to the Author):

The study describes the microfluidic approach of artificial cell-like polymersomes for the spatiotemporal control of signal-driven enzymatic reactions. Authors postulate that there are limited approaches for low-molecular weight substrates diffusion through membrane vesicles without the help of biopores etc. Especially, for the design and fabrication of giant unilamellar vesicles (GUV) by microfluidic approaches. This is true when only considering this specific technology for the design and fabrication of synthetic, polymeric GUVs. There is a need of further improvements for the design and fabrication of spatiotemporal control over enzymatic (cascade) reactions, while other approaches in the literature of controlled permeability of proteinosomes, polymersomes and membrane-stabilized coacervates etc. are still existing, leading to with and without integrated biopores, membrane proteins and channel proteins for specific molecule diffusions for carrying out spatiotemporal control over enzymatic reactions.

Authors have smoothly summarized the recent developments in microfluidic approaches, where all basic reactions (coacervate formation and actin polymerization) have been carried out in liposomal confinements, but also under reversible conditions for coacervate formation processes. Thus, the concept to carry out basic reactions in artificial cell compartments is a kind of standard within the community driven by a limited membrane permeability. In the introduction authors indicate the scientific challenge of their study that “However, elaborate machinery of external signal-driven enzymatic reaction or cascade reaction in artificial cells still remains unclear as the membrane permeability, stoichiometry of the reactants, and reaction kinetics, all needs to be carefully considered. Therefore, there is an unmet need for a new synthetic cellular model with in-depth understanding of the molecular transport property across the membrane for successfully implementation of enzymatic reactions.” When considering the achieved results of the study, then there is a limited progress in the direction of reversibility of coacervate formation within polymeric

GUV. Authors indicated that due to excess use of ions one excludes such experiments.

One interesting aspect of the study is the polymerization of G-actin via the formation of ATP-bound actin into the formation of F-actin cytoskeleton proven by fluorescence microscopy. It is true that the formation of actin filaments in GUV through the use polymeric vesicles is rarely described. Authors should cite following ref. for actin polymerization within polymersome-based GUV (Advanced Functional Materials (2020), 30(32), 2002949).

Generally, authors have smoothly described and well characterized their complex system (okay: statistics, error bars), including many control experiments carried out and experiments repeated many times. The methods used expect the standards within the research field.

The presented GUV exhibit an alternative approach for controlled encapsulation and controlled membrane-diffusion processes, but not such novelty in respect to spatiotemporal control over enzymatic reactions in combination with feedback-controlled reaction in dynamic or out-of equilibrium systems is not given. Authors present enzymatic (cascade) reactions without any further limitations, for example, through the change of temperature.

There are some points of improvements:

- Authors do not present any further information about the stability of polymeric GUV whether GUV are only working at RT, but also at 37°C. This point was not investigated along the stability whether there exists a RT stability for one week, but not at 37°C. In the experimental section there is no indication at which temperature enzymatic (cascade) reactions have been studied. What's about the stability of GUV at 37°C over a longer period (e.g. 7 days)
- Do the author give some more information about the encapsulation efficiency of enzymes and proteins in their GUV
- The experimental section of all experiments is not well presented. Authors are requested to write a complete protocol for each experiment in the SI. At the moment one cannot really follow what authors have really done. One concern is that master, PhD student or other scientists like to repeat such experiments, but there is no clear guide for the huge experiment series carried out by the authors. This is the weakest point of the manuscript.
- The discussion part is very weak, while there is no comparison with existing systems to outline which progress have been really done in the study.
- There is no real presence of experimental descriptions in SI for example for Fig. S4 and S10 as well. Authors are requested to do additional work.

Reviewer #3 (Remarks to the Author):

In this interesting article by Seo et al., the authors use capillary-based microfluidics to make polymersomes in a controlled manner and use these containers to study the formation of coacervate droplets through various triggers such as pH and enzymatic synthesis of components. They also couple component transport across the membrane to enzymatic activity and actin polymerization. The manuscript is clearly written, detailed, contains systematic analyses and the obtained results will be of interest to the synthetic biology community. I have several questions before the manuscript can be considered for publication.

Main comments

- The authors talk about the dewetting of cyclohexane droplets but the dewetting phenomenon is not shown anywhere. A video or a time-lapse image sequence would make it clear. Connected to this, I see small black dots on the polymersomes in fig. 2b1. Are these oil pockets? Does it then indicate that dewetting is not achieved completely?
- It is not clear how the osmotically induced swelling leads to the estimate of ion permeability. The authors mention the use of hypertonic solutions and identical osmolyte concentration in the same sentence (line 181), which is confusing.
- Given that the pH inside the polymersomes seems to equilibrate extremely fast (in less than a second, line 218) with the external environment, the coacervation process itself is extremely slow. Can the authors discuss this? It is not clear why coacervation needs more than an hour to go to completion and this needs to be clarified.

- Fig. 3a-b: It is not clear what the fitted lines are: are these polynomial fits or just a guide to the eye? Also, it is quite confusing that the authors have switched the colour codes for graph a and b. Accordingly, the labels for the two sketches in 3b are wrong. Also, the label 'single phase' is a bit confusing. Some simple modifications or addition of sketches could make this graph much more understandable.
- The authors have done a systematic and a detailed study of how ATP versus ADP is a better coacervation agent (section 2.3). However, this is a quite well-known fact in the field and in my opinion, not really a new addition. However, this is more of a side-remark, and I leave it up to the authors to modify that section or keep it as it is.
- It will be interesting to see if the formed coacervates can be dissolved by lowering the pH or through enzymatic means and will be a worthy addition to the manuscript.

Minor points

- The first sentence of the abstract: the part mentioning the cytoskeleton needs revising.
- Fig. 4b: the polymersomes can hardly be seen, please increase the contrast.
- Fig. 5b: the images are very faint. Also, it is hard to make out the different coloured circles in the graph.

REVIEWER COMMENTS

Reviewer #1 (Remarks to the Author):

This manuscript reports artificial cell-like polymersomes for controlled enzymatic reactions. The pluronic-based polymersomes were designed by microfluidics, of which membrane has a molecular cut-off threshold of permeation around 500 Da and allows the diffusion of metal ions. The authors encapsulated pyruvate kinase as well as ADP and PAH in the polymersomes. When the polymersomes were suspended in the solution of phosphoenol pyruvate (PEP), phosphoenol pyruvate diffuses through the membrane and the pyruvate kinase catalyzes the transfer of a phosphate group from PEP to ADP to yield ATP. As PAH and ATP produce a complex coacervate through electrostatic attraction, an inner compartment is formed inside the polymersome. Similarly, filamentous actin is produced through ATP-actin binding and polymerization when the polymersome contains globular actin. With encapsulation of pyruvate kinase and lactic dehydrogenase as well as ADP and NADH, a PEP-driven cascade enzymatic reaction is demonstrated. I read this manuscript with great interest and I believe that the polymersome in this work is promising as a platform for artificial cells. Therefore, I recommend the publication of this work in *Nature Communications* and hope the following minor comments help the authors improve their work:

We thank the reviewer's positive evaluation and constructive comments. As per reviewer's comments, one of the major aspects of this paper is that we utilize the selective molecular transmembrane permeation of Pluronic-based giant polymer vesicles without using any biopores to design in vitro enzymatic reactions in protocells. We hope that the revisions we are providing would further strengthen your positive evaluation of our work that it is worth publication in Nature Communications. Our point-by-point response to the reviewer #1's comments appear below:

1. In **Figures 1d and e**, is there no change in the size of the polymersome? The permeability might be overestimated if there was an inward flow of water due to imperfect osmolarity matching between the lumen and surrounding (hypotonic).

*We thank the reviewer for the helpful comments, and we agree that inclusion of additional molecules in the interior can lead to inward flow of water due to mismatch in osmolarity. However, in this set of experiment, the osmolality of polymersome interior and exterior was intentionally kept equal to prevent any size change. In particular, 10 wt% PEG (6 kDa) used as the polymersome interior is 100 ± 3 mOsm/kg while 10 wt% PVA (13-23 kDa) used as the external continuous phase corresponds to 103 ± 4 mOsm/kg. Therefore, the osmolyte concentrations inside and outside the polymersome prior to incorporation of fluorescent dye molecules were similar in value and are within the error range. Moreover, the concentration of all six fluorescent dyes used in the experiments were added in extremely low concentration (5 ppm), which do not noticeably alter the osmolyte concentration. Taken together, we believe that the volumetric change by influx of water due to imperfect osmolarity matching can be considered negligible in **Figure 1d and e**.*

To address the reviewer's comment, we have included a brief description of the fluorescent dye concentration in the revised manuscript as follows:

“Also, to determine the size-selective permeability of the membrane, we prepare sets of polymersomes each encapsulating six representative fluorescent dyes with different molecular weight (MW) and net charge in the aqueous core to monitor the retention of these dyes over time. We note that the polymersome interior and exterior are osmotically matched, 100 ± 3 and 103 ± 4 mOsm/kg, respectively, and fluorescent dye molecules are added in low concentrations (5 ppm) to minimize this difference. While numerous factors are known to affect the physical properties of the polymeric membrane, ...” (page 5, line 13-20)

2. In **Figure 1f**, it is suggested to discuss why the permeabilities of K^+ and Mg^{2+} ions are much smaller than those of small dyes. Are the hydrodynamic diameters of K^+ and Mg^{2+} ions larger due to hydration shells? Or is there any other molecular interaction to reject the ions?

We appreciate the reviewer's critical feedback. The estimated permeabilities of two ions were indeed relatively low compared to the values of the fluorescent dye molecules. As the reviewer pointed out, we believe that this is attributed to the difference in electric charge densities of the diffusing species. The two metal ions, K^+ and Mg^{2+} , have high electric charge density compared to the fluorescent molecules, which leads to larger hydrodynamic radius than that of the fluorescent molecules.[1, 2] In fact, it has been reported previously that these metal ions can strongly attract water molecules and exhibit thick hydration layer due to their high charge density in aqueous solution.[2] Likewise, the divalent cation Mg^{2+} exhibits thicker hydration layer and thus smaller value of permeability compared to K^+ .

As per the reviewer's comment, we have added a brief sentence as well as the relevant references in the revised manuscript to discuss the smaller value of permeabilities for K^+ and Mg^{2+} ions compared to the fluorescent dye molecules as follows:

“The estimated permeability values of K^+ and Mg^{2+} ions are small compared to the fluorescent dye molecules, possibly due to the high electric charge density which leads to larger hydrodynamic radius [1,2]. The permeability of these two ions are separately confirmed...” (page 6, line 17-23.)

List of references

- [1] A. Guskov & S. Eshaghi. *The Mechanisms of Mg^{2+} and Co^{2+} Transport by the CorA Family of Divalent Cation Transporters. Current Topics in Membranes. 69, p.399 – p.400 (2012)*
- [2] E. A. Permiakov. *Metalloproteomics. John Wiley & Sons. p. 786. (2009)*

3. In **Figure 2b**, is it necessary to increase pH up to 10 for the conservation? From pKa values of ATP and PAH, it seems that pH near 7-8 is appropriate to induce electrostatic attraction. Is there any influence of pH on the coacervate diameter and the time scale for coacervation? I also wonder whether the polymersomes remain stable for the range of pH from 4 – 10.

*We thank the reviewer for pointing out this important point. As the reviewer pointed out, the appropriate pH to induce complex coacervation is at the intermediate pH near 7-8 instead of pH 10. In the experiment shown in **Figure 2b**, the external pH is not adjusted to pH 10 but instead, aliquot amount of pH 10 buffered solution is introduced at the periphery of the polymersomes equilibrated at pH 4. Specifically, 5 mL of pH 10 buffer was introduced into 5 mL of polymersome solution maintained at pH 4 which leads to increase of the overall solution pH to 7.41 ± 0.05 . We thank again the reviewer for the critical comments, and we have revised the schematics of **Figure 2b** as well as the main text accordingly.*

(Revised Fig. 2) a) Schematics illustrating the complex coacervate formation between ATP and PAH. b) Schematics illustrating the pH-induced complex coacervates in polymersomes. i) Fluorescence micrograph showing no noticeable formation of complex coacervates within the polymersomes at pH

4. ii) Fluorescence micrograph showing complex coacervate formation within the polymersomes 45 min after adjusting the pH to 7.4. Both scale bars represent 100 μm . c) Plot showing the change in the number of coacervate droplets inside polymersomes after increasing the external pH condition to pH 7.4. (n=11). d) Plot showing the average diameter of the matured coacervate droplet (t=120 min) inside the polymersomes with variation in the ATP concentration (n=26). e) Series of magnified fluorescence micrographs showing the coacervation process from nucleation to maturation. Scale bar represents 100 μm . All error bars represent standard deviation.

“Upon increasing the external pH from 4 to 7.4 by introducing aliquot amount of pH 10 buffered solution near the periphery of polymersomes, we anticipated that ~~as the membrane exhibit high proton permeability,~~ ATP within the polymersomes would rapidly deprotonate and become negatively charged to complementarily interact with the positively charged PAH, resulting in complex coacervation. As expected, ~~we observe that~~ changing the external pH to 7.4 leads to nucleation of small coacervate droplets followed by maturation via aggregation of these coacervates into a single larger one, as evidenced by the dramatic change from uniform green fluorescence signal inside the polymersomes (Fig. 2b, i) to small coacervate droplets (Fig. 2b, ii) after 45 min.” (page 8, lines 27-34)

To further address the reviewer’s comment, we have performed additional experiments to determine the influence of pH on the resulting coacervate diameter as well as the time scale for coacervation. Similar to the previous experiment in which the overall solution pH was adjusted to pH 7.4, 8 mL of pH 10 buffered solution is introduced to the periphery of 2 mL polymersome solution at pH 4 to further increase the overall solution pH to 9.45 ± 0.02 . While we observe no significant difference in the time scale of coacervation, the diameter of the matured coacervate droplets formed inside polymersomes decreased from 27.4 ± 3.7 to $22.5 \pm 3.3 \mu\text{m}$ upon increasing the pH from 7.4 to 9.45, as shown in the newly prepared Supplementary Figure 12 below.

Supplementary Figure 12. a) Plot showing the change in the number of coacervate droplets inside polymersomes after increasing the pH condition to pH 9.45. (n=7). b) Plot showing the average diameter of the matured coacervate (t=120 min) inside the polymersomes with variation in the ATP concentration (n=5).

The reduction in the overall size of the matured coacervate droplet with increase in pH condition is possibly due to the dramatic reduction in the number fraction of charged amine groups (-NH³⁺) in PAH. As the electrostatic attraction between ATP and PAH leads to formation of complex coacervates, the pH change from 7.4 to 9.45 above the pK_a value of PAH substantially reduces the portion of protonated amine groups in PAH. The deprotonated (uncharged) portion of the PAH can be estimated using the equation below,

$$f = \frac{1}{1 + 10^{pH-pK_a}}$$

, where the f refers to the deprotonated (uncharged) portion of the PAH. We note that the equation above does not consider the salt concentrations and thus may be slightly different in our experimental conditions. However, quick evaluation of the f values at different pH conditions reveal that as the pH increases from 7.4 to 9.45 beyond the pK_a of PAH, the number fraction of the deprotonated amines increases from 2.5% to 83.8%. This indicates that the charged amine groups in PAH capable of forming complexation with ATP decreases, thereby resulting in matured coacervate droplet with smaller size.

*As per reviewer's comment, we have added **Supplementary Figure 12** in the Supplementary Information and added a brief description in the revised manuscript as follows:*

"We note that the dimension of the matured coacervate droplet depends on the ATP concentration (Fig. 2d) and exhibits an average diameter of 27.4 ± 3.7 μm at 10 mM ATP (Fig. 2e and Supplementary Fig. 11). Moreover, we find that the average diameter of the matured coacervate droplet decreases to 22.5 ± 3.3 μm when the overall solution pH is increased to pH 9.45 above the pK_a value of PAH due to the decrease in the charged amine groups in PAH capable of forming complex coacervates with ATP (Supplementary Fig. 12)." (page 9, lines 4-9)

Lastly, we also confirmed that the Pluronic-based polymersomes remained stable in all pH conditions mentioned from pH 4 to 9.45, as shown in the sets of optical micrographs below.

Figure. Sets of optical micrographs showing the intact Pluronic-based polymersomes at different pH conditions. Scale bar represents 100 μm .

4. It is suggested to increase the color contrast for symbols in Figure 5b.

*We appreciate the reviewer for the helpful comments. As per reviewer's comment, we have enhanced the color contrast for the symbols in **Revised Fig. 4b** to improve visibility.*

Reviewer #2 (Remarks to the Author):

The study describes the microfluidic approach of artificial cell-like polymersomes for the spatiotemporal control of signal-driven enzymatic reactions. Authors postulate that there are limited approaches for low-molecular weight substrates diffusion through membrane vesicles without the help of biopores etc. Especially, for the design and fabrication of giant unilamellar vesicles (GUV) by microfluidic approaches. This is true when only considering this specific technology for the design and fabrication of synthetic, polymeric GUVs. There is a need of further improvements for the design and fabrication of spatiotemporal control over enzymatic (cascade) reactions, while other approaches in the literature of controlled permeability of proteinosomes, polymersomes and membrane-stabilized coacervates etc. are still existing, leading to with and without integrated biopores, membrane proteins and channel proteins for specific molecule diffusions for carrying out spatiotemporal control over enzymatic reactions. Authors have smoothly summarized the recent developments in microfluidic approaches, where all basic reactions (coacervate formation and actin polymerization) have been carried out in liposomal confinements, but also under reversible conditions for coacervate formation processes. Thus, the concept to carry out basic reactions in artificial cell compartments is a kind of standard within the community driven by a limited membrane permeability. In the introduction authors indicate the scientific challenge of their study that “However, elaborate machinery of external signal-driven enzymatic reaction or cascade reaction in artificial cells still remains unclear as the membrane permeability, stoichiometry of the reactants, and reaction kinetics, all needs to be carefully considered. Therefore, there is an unmet need for a new synthetic cellular model with in-depth understanding of the molecular transport property across the membrane for successfully implementation of enzymatic reactions.” When considering the achieved results of the study, then there is a limited progress in the direction of reversibility of coacervate formation within polymeric GUV. Authors indicated that due to excess use of ions one excludes such experiments. One interesting aspect of the study is the polymerization of G-actin via the formation of ATP-bound actin into the formation of F-actin cytoskeleton proven by fluorescence microscopy. It is true that the formation of actin filaments in GUV through the use polymeric vesicles is rarely described. Authors should cite following ref. for actin polymerization within polymersome-based GUV (*Advanced Functional Materials* (2020), 30(32), 2002949).

Generally, authors have smoothly described and well characterized their complex system (okay: statistics, error bars), including many control experiments carried out and experiments repeated many times. The methods used expect the standards within the research field. The presented GUV exhibit an alternative approach for controlled encapsulation and controlled membrane-diffusion processes, but not such novelty in respect to spatiotemporal control over enzymatic reactions in combination with feedback-controlled reaction in dynamic or out-of equilibrium systems is not given. Authors present enzymatic (cascade) reactions without any further limitations, for example, through the change of temperature. There are some points of improvements:

We appreciate the positive evaluation and the constructive criticism expressed by the reviewer, and we agree that there has been considerable prior art investigating on the formation of coacervate droplets and actin polymerization in liposomes.[1-3] However, polymeric GUVs have been rarely investigated for spatiotemporal control of enzymatic reaction as well as formation of actin filaments other than the work that the reviewer pointed out.[4] This particular work that we have included in the manuscript is of critical importance as they showed for the first time that cytoskeletons can be formed within the polymeric GUV by signal-triggered cascade reaction. In addition, polymeric GUVs have clear advantage over lipid-based GUVs as they can remain stable up to 3 months [5] and are stable in broad range of pH conditions as shown in the figure below.

Figure. Sets of optical micrographs showing the intact Pluronic-based polymersomes at different pH conditions. Scale bar represents 100 μm .

More importantly, among various approaches to prepare GUVs, microfluidic synthesis of polymersomes enable precise control over the size and composition, providing a firm basis for spatiotemporally regulating biochemical reactions in artificial cell-like vesicles.

In this work, we focus not just on the microfluidic fabrication of polymersomes but instead on the design of stable and semi-permeable membrane without usage of biopores. This loosely knitted bilayer membrane allows efficient encapsulation of enzymes while external signals smaller than the molecular weight cut-off value to readily permeate through the membrane, inducing enzymatic reaction and successive coacervation or actin polymerization in these polymersomes. Therefore, we believe that the current work does represent a considerable advance over the work done previously.

Nevertheless, we have responded to all the comments that the reviewer provided and performed additional experiments to strengthen our results and to further convince the readers of the importance and originality of this work. We thank the reviewer for the helpful comments and suggestions, which have substantially improved the clarity of our manuscript. We hope that with the additional experiments and the revisions that we are providing, the reviewer will

now agree with reviewer #1 who stated that “recommend the publication of this work in *Nature Communications*”. Our point-by-point response to the comments of reviewer #2 appear below:

List of references

- [1] S. Deshpande, F. Brandenburg, A. Lau, M. G. F. Last, W. K. Spoelstra, L. Reese, S. Wunnava, M. Dogterom & C. Dekker. *Spatiotemporal control of coacervate formation within liposomes*. *Nat. Commun.* **10**, 1800 (2019)
- [2] C. Love, J. Steinkühler, D. T. Gonzales, N. Yandrapalli, T. Robinson, R. Dimova & T.-Y. D. Tang. *Reversible pH-Responsive Coacervate Formation in Lipid Vesicles Activates Dormant Enzymatic Reactions*. *Angew. Chem. Int. Ed.* **132**, 6006-6013 (2020)
- [3] L.-L. Pontani, J. van der Gucht, G. Salbreuc, J. Heuvingh, J.-F. Joanny & C. Sykes. *Reconstitution of an actin cortex inside a liposome*. *Biophys. J.* **96**, 192-198 (2009)
- [4] A. Belluati, S. Thamboo, A. Najer, V. Maffei, C. von Planta, I. Craciun, C. G. Palivan & W. Meier. *Multicompartment Polymer Vesicles with Artificial Organelles for Signal-Triggered Cascade Reactions Including Cytoskeleton Formation*. *Adv. Funct. Mater.* **30**, 2002949 (2020)
- [5] E. Rideau, R. Dimova, P. Schwille, F. R. Wurm & K. Landfester. *Liposomes and polymersomes: a comparative review towards cell mimicking*. *Chem. Soc. Rev.* **47**, 8572-8610 (2018)

-Authors do not present any further information about the stability of polymeric GUV whether GUV are only working at RT, but also at 37°C. This point was not investigated along the stability whether there exists a RT stability for one week, but not at 37°C. In the experimental section there is no indication at which temperature enzymatic (cascade) reactions have been studied. What's about the stability of GUV at 37°C over a longer period (e.g. 7 days)?

We thank the reviewer for the valuable feedback. All our studies on complex coacervation and enzymatic reactions were performed at RT. As per the request of the reviewer, the temperature conditions for all the experiments were noted in the Methods section. In addition, we observed that the polymersomes remain stable and retain their structure even after 1 week of incubation in a temperature-adjustable chamber set at 37.0 ± 0.5 °C, as shown in the optical (left) and fluorescence micrograph (right) below.

Supplementary Figure 4. Optical and fluorescence micrograph of the Pluronic-based polymersomes after 1 week of incubation in a temperature-adjustable chamber set at 37.0 ± 0.5 °C. Scale bar represents 100 μm .

As per reviewer's suggestion, we have added the newly prepared **Supplementary Figure 4** in the Supplementary Information and the relevant sentences on the thermal stability of the polymersomes in the revised manuscript.

“Unlike the conventional lipid-based giant unilamellar vesicles, the resulting Pluronic-based polymersomes are stable for at least one week at room temperature as well as 37 °C (Supplementary Fig. 3 and 4) and monodisperse in size with mean diameter of 118 μm and coefficient of variation (CV) of 4.0 % (Supplementary Fig. 5).” (page 5, line 10-13)

- Do the author give some more information about the encapsulation efficiency of enzymes and proteins in their GUV?

We thank the reviewer for the helpful comments, and we agree with the reviewer that more information about the encapsulation efficiency of enzymes and proteins in GUV would be useful. However, while the encapsulation efficiency of fluorescent dye-labeled enzymes and proteins can be precisely determined using fluorescence correlation spectroscopy (FCS) as reported by others,[1] this is unnecessary if to use microfluidic approaches to prepare polymeric GUVs. In fact, one of the key merits of using this specific approach over others is that one can prepare monodisperse emulsions and emulsion-templated vesicles with fine tunable size and composition in a continuous fashion with nearly 100% encapsulation efficiency.[1, 2]

Nevertheless, we have conducted a new experiment in which we labeled pyruvate kinase with a green fluorescent dye, fluorescein, following the Biotium's protocol with N-Hydroxysuccinimide (NHS)-fluorescein.[3] Then, we incorporated the fluorescein-labeled pyruvate kinase in polymersomes and monitored the change in fluorescence intensity using a confocal microscope (Leica, Stellaris). We observe no noticeable fluorescence signal outside the polymersomes after collection as well as decrease in the signal at the interior of the

polymersomes as shown in the **Supplementary Figure S21** below. This clearly shows that the pyruvate kinase is encapsulated with negligible loss during preparation and that they are well-retained within the polymersomes.

Supplementary Figure 21. (a) Fluorescence micrograph showing the Pluronic-based polymersomes encapsulating Fluorescein labeled pyruvate kinase. Scale bar represents 100 μm. (b) Plot represents the spatially resolved intensity profile across the polymersome interior and exterior. (c) Plot and fluorescence micrographs showing the steady fluorescence intensity within the polymersome. Scale bar represents 100 μm.

As per reviewer's comment, we have added the newly prepared **Supplementary Figure 21** and the relevant descriptions in the Supplementary Information as well as the revised manuscript.

"We note that PyK is encapsulated with negligible loss and well-retained within the polymersomes (Supplementary Fig. 21) and that equal concentration of ADP was added in the aqueous continuous phase during polymersome production as well as in the collection bath to prevent the diffusion of ADP (MW 427 Da) across the membrane." (page 12, lines 25-28)

Note 1. Fluorescence Micrograph Acquisition and FITC-PyK Synthesis. Fluorescence micrograph was acquired from a confocal microscope (STELLARIS 5, Leica). The spatially resolved intensity plot was obtained using ImageJ. To synthesize fluorescein labeled pyruvate kinase (FITC-PyK), we first prepare a solution of 2.5 mg mL⁻¹ of PyK in 0.1 M sodium buffer solution (pH 8.5). Then, the dye stock solution was separately prepared by dissolving 10 mM NHS-Fluorescein in anhydrous DMSO. Next, 25 μL of dye stock solution was drop-wise injected into a vial containing 1 mL of PyK stock solution and gently stirred at 600 rpm in the dark (24 °C) for at least 3 h. To remove the unreacted dye, we use a PD-10 column (Cytiva) equilibrated with 0.1 M HEPES buffer. After removal of the unreacted dye by collecting the first band, we dialyzed overnight at room temperature using Slide-A-Lyzer Dialysis Cassette, 10K MWCO (ThermoFisher) for further purification." (page 20, lines 7-16, Supplementary Information)

List of references

[1] dos Santos, E. C., Belluati, A., Necula, D., Scherrer, D., Meyer, C. E., Wehr, R. P., Lörtscher, E., Palivan, C. G. & Meier, W. Combinatorial Strategy for Studying Biochemical Pathways in Double Emulsion Templated Cell-Sized Compartments. *Adv. Mater.* **32**, 2004804 (2020)

[2] Meyer, C. E., Abran, S.-L., Craciun, I. & Palivan, C. G. Biomolecule-polymer hybrid compartments: combining the best of both worlds. *Phys. Chem. Chem. Phys.* **22**, 11197-11218 (2020)

[3] Protocol: Succinimidyl Ester Labeling of Protein Amines, Biotium, (2020) URL: <https://biotium.com/tech-tips/protocol-succinimidyl-ester-labeling-of-protein-amines/>

- The experimental section of all experiments is not well presented. Authors are requested to write a complete protocol for each experiment in the SI. At the moment one cannot really follow what authors have really done. One concern is that master, PhD student or other scientists like to repeat such experiments, but there is no clear guide for the huge experiment series carried out by the authors. This is the weakest point of the manuscript.

We appreciate the reviewer's critical feedback, and we agree with the reviewer that the previous version of the manuscript did not sufficiently describe the protocols used for each experiment. As per reviewer's suggestion, we have revised the manuscript and provided the complete protocol in the Supplementary Information. The detailed protocols that have been newly added to the revised manuscript and Supplementary Information appears below.

"To estimate the ion permeability, we adopt the membrane transport equation to describe the ion flux through a thin polymersome membrane as follows.[1-4]

$$\frac{dC_{ions}}{dt} = \frac{P * A}{V} * \Delta C \dots (1)$$

,where C is concentration, P is permeability, A is surface area, and V is volume. By multiplying both sides by V, equation (1) can be re-written in terms of total number of ions. ..." (page 2, Supplementary Information)

"To fabricate these lipid-based giant unilamellar vesicles, we use a separate glass capillary-based microfluidic device which comprises of a square capillary and two tapered cylindrical capillaries, to form W/O/W double-emulsion template..." (page 5, lines 4-17, Supplementary Information)

*"Acquisition of the polymersome swelling upon inserting 0.1 mg mL⁻¹ of K⁺ ion-selective channel, valinomycin, into the polymersomes prior to incubation in the hypertonic KCl solution confirms that the polymersomes are indeed permeable to K⁺ and that the permeability can be enhanced by approximately 2.67-folds (P_{K⁺} = 1.74 nm s⁻¹) via inclusion of ion-channels as shown in **Supplementary Fig. 7a.** ..."* (page 8, lines 7-17, Supplementary Information)

“To check proton permeability, we prepare a separate set of polymersomes encapsulating 5 $\mu\text{g mL}^{-1}$ HPTS solution adjusted to acidic pH of 4.0. After polymersomes are settle in the collection bath, we gently add an aliquot amount of pH 10.0 buffered solution to the periphery of polymersomes to increase the overall pH condition to 7.4.” (page 9, lines 5-8, Supplementary Information)

*“To verify whether ATP (MW 507 Da) can be retained within the membrane, we introduce ATP outside the polymersomes containing only PAH in the aqueous core. We find that even when the polymersomes containing an aqueous solution of 10 mg mL^{-1} PAH and 0.5 mg mL^{-1} FITC-PAH (pH 4) are subjected to pH 8 adjusted media containing 10 mM ATP solution for 24 h, no noticeable formation of complex coacervate is observed, as evidenced by the homogeneous green fluorescence signal within the polymersomes (**Supplementary Fig. 9**). Consistent with the membrane MW cut-off acquired previously, this indicates that ATP cannot effectively diffuse through the polymersomes’ membrane.” (page 10, lines 6-12, Supplementary Information)*

“To verify the pH-controllable complex coacervation between ATP and PAH, we prepare an aqueous solution containing 10 mM ATP and 10 mg mL^{-1} of PAH (MW \sim 17 500 Da). Additionally, 0.5 mg mL^{-1} of green fluorescent dye labeled PAH (poly(fluorescein isothiocyanate allylamine hydrochloride, FITC-PAH) was added for visualization purposes. We observe no apparent coacervate formation at pH 4 and pH 11, which is either lower than the pK_a of ATP or higher than the pK_a of PAH. However, at the intermediate pH of 7.4 at which both ATP and PAH are oppositely charged, complex coacervate forms within a minute.” (page 11, lines 8-14, Supplementary Information)

*“Through **Supplementary Fig. S15-S17**, we conducted a separate set of experiment in which two sets polymersomes each containing i) 10 mM ATP, 5 mg mL^{-1} PAH, and ii) 5 mM ATP, 10 mg mL^{-1} PAH, respectively, in the aqueous core were prepared to further verify whether the surface charge of the complex coacervate as well as the direction of pH transition indeed affects the time scale of coacervation...” (page 15, lines 6-25, Supplementary Information)*

*“We investigate the extent of coacervation at pH 7.4 buffer solution in bulk for both nucleotides, ADP and ATP, with respect to PAH using a turbidity assay. We acquire the turbidity value while we fix the PAH concentration at 10 mg mL^{-1} and vary the concentrations of nucleotides from 0 to 20 mM (**Supplementary Fig. 18a**). Here, relatively high concentration of PAH (10 mg mL^{-1}) was used for the analysis of coacervate formation,^[8, 43] as their behavior at lower concentration regime is sensitive and difficult to quantify using the turbidity assay, as reported previously by others ...” (page 17, lines 9-24, and page 18, lines 1-24, Supplementary Information)*

“Fluorescence micrograph was acquired from a confocal microscope (STELLARIS 5, Leica). The spatially resolved intensity plot was obtained using ImageJ. ...” (page 20, lines 7-22, Supplementary Information)

“We use Bradford assay to calculate the concentration of synthesized FITC-PyK, as previously reported by others and described in the manufacturer’s protocol. ...” (page 20, lines 23-30, and page 21, lines 1-8, Supplementary Information)

“To investigate the enzymatically induced coacervation process in polymersomes, we measure the number of coacervate droplets and their dimensions after PEP infusion as a function of time. ...” (page 22, lines 5-9, Supplementary Information)

“Actin polymerization process is commenced by injection of PEP, K^+ , and Mg^{2+} at the periphery of polymersomes whose membrane allow these molecules to diffuse into the lumen. ...” (page 24, lines 1-7, Supplementary Information)

We hope that with the detailed protocol that we are providing, the readers can now repeat any experiments whenever necessary.

- The discussion part is very weak, while there is no comparison with existing systems to outline which progress have been really done in the study.

We much appreciate the reviewer for the critical feedback and agree that our Pluronic-based polymersomes have not been properly compared with existing systems to highlight its unique set of properties. As discussed previously, there have been considerable prior work on liposomes for spatiotemporal control of enzymatic reactions where coacervate formation and actin polymerization have been carried out in these liposomal confinements. However, the size and the composition of the lipid-based vesicles are difficult to control and thus the initial condition as well as the onset point of biochemical reaction is hard to regulate. While these limitations can be mitigated to some extent by utilizing microfluidic approaches, the resulting lipid-based giant unilamellar vesicles (GUVs) are not as stable as polymeric GUVs. This limits the broader usage of these vesicles in the bottom-up approach to design responsive and programmable artificial cell-like systems.

*To directly address the reviewer’s comment and verify the enhanced stability of these polymeric GUVs, we have conducted a new experiment in which lipid-based GUVs were prepared for comparison with the Pluronic-based polymersomes. Briefly, the amphiphilic triblock copolymer, Pluronic L121, which was used as the building block was replaced with dipalmitoylphosphatidylcholine (DOPC). 5 mg mL⁻¹ of DOPC was dissolved in a mixture of chloroform and cyclohexane (36:64 vol%) and the size of the resulting lipid-based GUV was tuned to match the dimension of the polymersomes. We note that 0.5 wt% of Poloxamer surfactant (P188) was additionally dissolved in the continuous phase in the case of lipid-based GUVs to facilitate the dewetting of the oil droplet[1] and enhance the stability.[2-3] Nevertheless, lipid-based GUVs prepared using P188 were relatively unstable compared to that of polymersomes, similar to what observed by others.[2] We observe that they cannot stably maintain the vesicular structure for more than a day as shown in **Supplementary Figure 3** below.*

Supplementary Figure 3. Series of optical micrographs showing the dewetting transition and rupture over time for lipid-based giant unilamellar vesicles (GUVs). Scale bar represents 100 μm .

This indicates that they are not well-suited for spatiotemporal control of biochemical reactions since simple procedure such as exchange of buffer solution can lead to rupture of the vesicles, which can alter the concentration of the constituents in the surrounding environment and compromise the experiment. On the contrary, the Pluronic L121-based polymersomes presented in this work are stable and retain their structure in broad range of pH conditions and remain intact even after 1 week of incubation in a temperature-adjustable chamber set at 37.0 ± 0.5 °C, enabling extension of these artificial cell-like vesicles to other enzymatic reactions and external signals to impart additional functionality.

*As per reviewer's comment, we have added the newly prepared **Supplementary Figure 3** in the Supplementary Information and added a brief description in the revised manuscript to highlight the stability of Pluronic-based polymersomes as follows:*

"Unlike the conventional lipid-based giant unilamellar vesicles (GUVs), the resulting Pluronic-based polymersomes are stable for at least one week at room temperature as well as 37°C (Supplementary Fig. 3 and 4) and monodisperse in size with mean diameter of 118 μm and coefficient of variation (CV) of 4.0 % (Supplementary Fig. 5)" (page 5, line 10-13)

List of references

[1] N.-N. Deng, M. Yelleswarapu & W. T. S. Huck. Monodisperse Uni- and Multicompartment Liposomes. *J. Am. Chem. Soc.* **138**, 7584-7591 (2016)

[2] L. R. Arriaga, S. S. Datta, S.-H. Kim, E. Amstad, T. E. Kodger, F. Monroy & D. A. Weitz. Ultrathin Shell Double Emulsion Templated Giant Unilamellar Lipid Vesicles with Controlled Microdomain Formation. *Small* **10**, 950-56 (2014)

[3] E. A. Frankel, P. C. Bevilacqua & C. D. Keating. Polyamine/Nucleotide Coacervates Provide Strong Compartmentalization of Mg^{2+} , Nucleotides, and RNA. *Langmuir* **32**, 2041–2049 (2016)

- There is no real presence of experimental descriptions in SI for example for Fig. S4 and S10 as well. Authors are requested to do additional work.

We thank the reviewer for the helpful suggestion, and we also agree that experimental descriptions for the abovementioned figures in the Supplementary Information are either lacking or incomplete. To directly address the reviewer's comment, we have added the detailed experimental descriptions in the methods section of the revised manuscript as well as Supplementary Information. These revised descriptions appear as follows:

“To investigate the enzymatically induced coacervation process in polymersomes, we measure the number of coacervate droplets and their dimensions after PEP infusion as a function of time. Coacervation process is commenced by injection PEP at the periphery of polymersomes whose membrane allow PEP diffusion into the lumen. The formed coacervate droplets gradually merge into a single large droplet approximately after 120 min and increase up to $20.9 \pm 1.8 \mu\text{m}$ in diameter after 3 h.” (page 22, lines 5-9, Supplementary Information)

*“Actin polymerization process is commenced by injection of PEP, K^+ , and Mg^{2+} at the periphery of polymersomes whose membrane allow these molecules to diffuse into the lumen. To observe the actin polymerization shown in **Supplementary Fig. 25** and **26**, we use a confocal microscope (STELLARIS 5 Confocal Microscope, Leica) equipped with a HC PL 10X and 20X Plan apochromatic objective. Confocal micrographs were obtained using 488 nm OPSL CW laser, and 561 nm DPSS laser lines to excite Alexa Fluor 488 Phalloidin and Nile Red respectively, and were analyzed using a LAS X Software (Leica).” (page 24, lines 1-7, Supplementary Information)*

Reviewer #3 (Remarks to the Author):

In this interesting article by Seo et al., the authors use capillary-based microfluidics to make polymersomes in a controlled manner and use these containers to study the formation of coacervate droplets through various triggers such as pH and enzymatic synthesis of components. They also couple component transport across the membrane to enzymatic activity and actin polymerization. The manuscript is clearly written, detailed, contains systematic analyses and the obtained results will be of interest to the synthetic biology community. I have several questions before the manuscript can be considered for publication.

We much appreciate the reviewer's positive evaluation and constructive comments. As per reviewer's suggestions, we have performed additional experiments and revisions to directly respond to the questions raised by the reviewer. Our point-by-point response to the comments and suggestions of the reviewer appear below.

Main comments

- The authors talk about the dewetting of cyclohexane droplets but the dewetting phenomenon is not shown anywhere. A video or a time-lapse image sequence would make it clear. Connected to this, I see small black dots on the polymersomes in fig. 2b1. Are these oil pockets? Does it then indicate that dewetting is not achieved completely?

We thank the reviewer for bringing up this important point. Unlike several other systems where the poor solvent droplet fully detaches from the vesicle surface, cyclohexane droplet partially dewets from the Pluronic-based polymersome and remain attached in the form of oil pocket until it thins out by evaporation.

*To directly address the reviewer's comment, we prepared a new set of Pluronic-based polymersomes and monitored the dewetting phenomenon over time as shown in **Supplementary Figure 2** below.*

Supplementary Figure 2. Series of optical micrographs showing the dewetting transition of Pluronic-based polymersomes. Scale bar represents 50 μm .

As cyclohexane exhibit lower density than water, the cyclohexane droplet remains centered towards the top, followed by evaporation to form a polymer bilayer. Similar phenomenon involving oil pockets has been observed by others for Pluronic L121-based vesicles and has been attributed to the absence of additional surfactant and the low hydrophobicity of the

building block. [2]. However, it was also shown that the polymeric bilayer formation can be facilitated by utilizing ultrathin-shell double emulsion templates as well as a large container filled with excess amount of aqueous solution to avoid saturation of the aqueous continuous phase with chloroform that may impede the rate of evaporation.[3] Indeed, we find that this leads to formation of polymeric bilayer throughout most of the vesicle within few minutes after collection. Moreover, in all our diffusion study, we do not observe any noticeable difference in the permeability value, indicating that the residual oil pocket does not interfere with the interpretation of the experimental data. We also note that the presence of residual oil pocket has been also observed by others in liposomes and their influence on subsequent enzymatic reactions as well as insertion of membrane pore protein (e.g., alpha-hemolysin) has been reported to be negligible.[1]

As for the small black dots in **Fig. 2b**, these are not oil pockets but an artifact by the mobility of the polymersome during confocal imaging.

These new results appear as **Supplementary Figure 2** in the Supplementary Information and a brief description has been also added in the revised manuscript as follows:

“followed by dewetting of the cyclohexane droplet within few minutes to form a polymer bilayer comprising of Pluronic L121, as illustrated in the schematics of the Pluronic-based polymersomes in **Figure 1b** and **Supplementary Fig. 2.**” (page 5, lines 8-10)

List of references

[1] L. R. Arriaga, S. S. Datta, S.-H. Kim, E. Amstad, T. E. Kodger, F. Monroy & D. A. Weitz. Ultrathin Shell Double Emulsion Templated Giant Unilamellar Lipid Vesicles with Controlled Microdomain Formation. *Small* **10**, 950-56 (2014)

[2] D. F. do Nascimento, L. R. Arriaga, M. Eggersdorfer, R. Ziblat, M. F. V. Marques, F. Reynaud, S. A. Koehler & D. A. Weitz. Microfluidic Fabrication of Pluronic Vesicles with Controlled Permeability. *Langmuir* **32**, 5350-5355 (2016)

[3] R. Luo, K. Göpflich, I. Platzman & J. P. Spatz. DNA-Nasende Assembly of Multi-Compartment Polymersome Networks. *Adv. Funct. Mater.* **30**, 2003480 (2020).

• It is not clear how the osmotically induced swelling leads to the estimate of ion permeability. The authors mention the use of hypertonic solutions and identical osmolyte concentration in the same sentence (line 181), which is confusing.

We thank the reviewer for the helpful comments, and we agree that the sentence that the reviewer refers to is indeed confusing. What we meant here by “identical osmolyte concentrations” is that the hypertonic solutions comprising of either 3.72 wt% KCl or 3.12 wt% MgCl₂ are similar in osmolyte concentrations with values of 888 ± 5 and 897 ± 3 mOsm/kg, respectively. We also agree that the method [1] employed to estimate the ion permeability is quite complex and require additional explanation for the readers to fully understand this method.

To determine the ion permeability, we employed the method proposed by Nascimento et al.[1] in which osmotically balanced polymersomes containing 10wt% PEG (6 kDa, 100 ± 3 mOsm/kg) solution in the interior are first prepared in 10 wt% PVA (13-23 kDa, 103 ± 4 mOsm/kg) solution. Then, aliquot amount of this polymersome dispersion is introduced into a large container filled with excess amount of hypertonic solution comprising of either 3.72 wt% KCl or 3.12 wt% MgCl₂ with osmolyte concentrations of 888 ± 5 and 897 ± 3 mOsm/kg, respectively. Since these ions are smaller than the MW cut-off value (500 Da) of the polymersome membrane, ions in the exterior diffuse through the membrane into the interior even though the osmolyte concentration of the exterior is initially higher than the interior. The influx of ions into the polymersome via diffusion reduces the ion concentration difference across the membrane, leading to inflow of water due to the presence of 10 wt% PEG in the interior that is larger than the MW cut-off. Therefore, the volumetric swelling of polymersome under such condition can be attributed to the ion permeation through the polymersome membrane.

To estimate the ion permeability, we adopt the membrane transport equation to describe the ion flux through a thin polymersome membrane as follows.[2-4]

$$\frac{dC_{ions}}{dt} = \frac{P * A}{V} * \Delta C \dots (1)$$

,where C is concentration, P is permeability, A is surface area, and V is volume. By multiplying both sides by V , equation (1) can be re-written in terms of total number of ions.

$$\frac{dN_{ions}}{dt} = P * A * \Delta C \dots (2)$$

As the influx of water molecules through the membrane is much faster than the permeability of ions, ion permeation can be regarded as the rate-determining step in the overall process. Accordingly, the equation of water influx into the polymersome can be replaced with the following equation (3) and (4) which describe the instantaneous osmotic equilibrium of the polymersome due to difference in osmolality.[3]

$$C_{ions}^{outer} = \frac{n_{ions}^{vesicles} + n_{PEG}^{vesicles}}{V^{vesicles}} \dots (3)$$

$$C_{ions}^{outer} * V^{vesicles} = n_{ions}^{vesicles} + n_{PEG}^{vesicles} \dots (4)$$

Next, both sides of equation (4) are differentiated with respect to time. Since the volume of the outer continuous phase is remarkably large compared to the volume within the polymersomes, we assumed that the initial concentration of the ions would be constant. Accordingly, following equation (5) can be obtained.

$$C_{ions}^{eq} * \frac{dV^{vesicles}}{dt} = \frac{dn_{ions}^{vesicles}}{dt} \dots (5) (\because C_{ions}^{outer} \sim C_{ions}^{eq}, n_{PEG}^{vesicles} = const)$$

Substituting the right-hand side of equation (5) with equation (2) yields the following expressions.

$$C_{ions}^{eq} * \frac{dV^{vesicles}}{dt} = P * A * \Delta C \dots (6)$$

$$\frac{dV^{vesicles}}{dt} = \frac{P * A * \Delta C}{C_{ions}^{eq}} = \frac{P * A}{C_{ions}^{eq}} * \frac{n_{PEG}^{vesicles}}{V} \dots (7)$$

$$4\pi R^2 * \frac{dR}{dt} = P * \frac{4\pi R^2}{C_{ions}^{eq}} * \frac{n_{PEG}^{vesicles}}{\frac{4\pi R^3}{3}} \dots (8)$$

$$\frac{dR}{dt} = 3 * \frac{P_{ions}}{C_{ions}^{eq}} * \frac{n_{PEG}^{vesicles}}{4\pi R^3} \dots (9)$$

Integrating both sides with respect to time allows acquisition of the final equation (10) below which relates the ion permeability to normalized polymersome radius change with respect to time.

$$\left(\frac{R}{R_0}\right)^4 = 1 + \frac{4 * P_{ions}}{R_0} * t \dots (10)$$

To improve clarity, we have revised the relevant descriptions in the manuscript and added the abovementioned derivations in the Supplementary Information as follows:

“As Pluronic-based polymersomes exhibit a MW cut-off value of 500 Da, encapsulation of 10 wt% PEG (MW 6,000 Da) in the aqueous core of polymersomes, and subsequent exposure to hypertonic solutions comprising of ions smaller than the membrane MW cut-off allows determination of the permeability of K^+ and Mg^{2+} through the membrane. Here, hypertonic solution comprising of either 3.72 wt% KCl or 3.12 wt% $MgCl_2$ solutions with osmolyte concentration of 888 ± 5 and 897 ± 3 mOsm kg^{-1} , respectively, were used as the hypertonic solutions.” (page 6, lines 8-14)

“Detailed derivation for determination of ion permeability. To estimate the ion permeability, we adopt the membrane transport equation to describe the ion flux through a thin polymersome membrane as follows.[1-4]

$$\frac{dC_{ions}}{dt} = \frac{P * A}{V} * \Delta C \dots (1)$$

,where C is concentration, P is permeability, A is surface area, and V is volume. By multiplying both sides by V , equation (1) can be re-written in terms of total number of ions.

$$\frac{dN_{ions}}{dt} = P * A * \Delta C \dots (2)$$

As the influx of water molecules through the membrane is much faster than the permeability of ions, ion permeation can be regarded as the rate-determining step in the overall process. Accordingly, the equation for the water influx to the polymersome can be replaced with the following equation (3) and (4) which describe the instantaneous osmotic equilibrium of the polymersome due to difference in osmolality.[3]

$$C_{ions}^{outer} = \frac{n_{ions}^{vesicles} + n_{PEG}^{vesicles}}{V^{vesicles}} \dots (3)$$

$$C_{ions}^{outer} * V^{vesicles} = n_{ions}^{vesicles} + n_{PEG}^{vesicles} \dots (4)$$

Consequently, both sides of equation (4) can be differentiated with respect to time. Since the volume of the outer continuous phase is remarkably large compared to the volume within the polymersomes, we assumed that the initial concentration of the ions would be constant. Accordingly, following equation (5) can be obtained.

$$C_{ions}^{eq} * \frac{dV^{vesicles}}{dt} = \frac{dn_{ions}^{vesicles}}{dt} \dots (5) (\because C_{ions}^{outer} \sim C_{ions}^{eq}, n_{PEG}^{vesicles} = const)$$

Substituting the right-hand side of equation (5) with equation (2) yields the following expressions.

$$C_{ions}^{eq} * \frac{dV^{vesicles}}{dt} = P * A * \Delta C \dots (6)$$

$$\frac{dV^{vesicles}}{dt} = \frac{P * A * \Delta C}{C_{ions}^{eq}} = \frac{P * A}{C_{ions}^{eq}} * \frac{n_{PEG}^{vesicles}}{V} \dots (7)$$

$$4\pi R^2 * \frac{dR}{dt} = P * \frac{4\pi R^2}{C_{ions}^{eq}} * \frac{n_{PEG}^{vesicles}}{\frac{4\pi R^3}{3}} \dots (8)$$

$$\frac{dR}{dt} = 3 * \frac{P_{ions}}{C_{ions}^{eq}} * \frac{n_{PEG}^{vesicles}}{4\pi R^3} \dots (9)$$

Integrating both sides with respect to time allows acquisition of the final equation (10) below which relates the ion permeability to normalized polymersome radius change with respect to time.

$$\left(\frac{R}{R_0}\right)^4 = 1 + \frac{4 * P_{ions}}{R_0} * t \dots (10)''$$

(page 2, Supplementary Information)

List of references

- [1] D. F. do Nascimento, L. R. Arriaga, M. Eggersdorfer, R. Ziblat, M. F. V. Marques, F. Reynaud, S. A. Koehler & D. A. Weitz. Microfluidic Fabrication of Pluronic Vesicles with Controlled Permeability *Langmuir* **32**, 5350-5355 (2016)
- [2] A. R. Allnatt. Theory of Phenomenological Coefficients in Solid-State Diffusion. I. General Expressions *J. Chem. Phys.* **43**, 1855 (1965)
- [3] M. Chabanon, J. C. S. Ho, B. Liedberg, A. N. Parikh & P. Rangamani. Pulsatile Lipid Vesicles under Osmotic Stress. *Biophys. J.* **112**, 1682-1691 (2017)
- [4] O. Kedem & A. Katchalsky. A Physical Interpretation of the Phenomenological Coefficients of Membrane Permeability. *J. Gen. Physiol.* **45**, 143-170 (1961)

- Given that the pH inside the polymersomes seems to equilibrate extremely fast (in less than a second, line 218) with the external environment, the coacervation process itself is extremely slow. Can the authors discuss this? It is not clear why coacervation needs more than an hour to go to completion and this needs to be clarified.

*We appreciate the reviewer's critical feedback. As evidenced by the rapid increase in fluorescence intensity within the polymersome shown in **Supplementary Figure 8**, polymersomes exhibit high proton permeability and thus it is not the rate-determining step in pH-induced complex coacervation.*

To further verify the origin of this large discrepancy in time scale of proton diffusion and pH-induced complex coacervation, we have performed extensive additional experiments in which we investigated the effect of overall polymersome size and the electrostatic interaction among PAH and ATP as well as coacervate droplets.

*As the coacervation occurs through nucleation of small coacervate droplets followed by maturation into a single larger one, we first studied how the polymersome size affects the time scale of complex coacervation with the assumption that reducing the size of the polymersome would decrease the time scale since the 3D-diffusion length during maturation process becomes shorter. To achieve this, we prepared sets of polymersomes with identical composition but with different overall sizes and induced complex coacervation by increasing the pH from 4 to 7.4. We observe that the time scale of maturation into a large coacervate droplet decreased by about 20 min and the matured coacervate droplet diameter decreased from 27.4 ± 3.7 to $23.3 \pm 2.0 \mu\text{m}$ when the polymersome size was reduced from 118 to $78.91 \pm 4.54 \mu\text{m}$ (**Supplementary Figure 14**).*

Supplementary Figure 14. a) Plot showing the change in the number of coacervate droplets inside smaller-sized polymersomes (10 mM ATP, 10 mg mL⁻¹ PAH) with diameter of $78.91 \pm 4.54 \mu\text{m}$ after increasing the pH condition to pH 7.4. (n=12). b) Plot showing the change in the diameter of the matured coacervate inside these polymersomes with time (n=5).

Next, we studied how the electrostatic interaction between PAH and ATP affects the overall coacervation process. Zeta-potential value of the matured coacervate droplet formed in bulk for 10 mM ATP and 10 mg mL⁻¹ PAH was $+61.1 \pm 3.8$ mV, indicating that there exists excess amount of positively charged PAH compared to ATP with protonated amine groups presented toward the coacervate surface.[1] Based on this observation, we investigated whether the presence of excess amount of protonated PAH initially affects the electrostatic interaction with ATP as well as maturation into a single large one. To do this, we induced complex coacervation in the direction of lowering the pH from pH 10.5 to pH 7.4, as opposed to the direction of increasing the pH from pH 4 to pH 7.4.

Supplementary Figure 15. a) Plot showing the change in the number of coacervate droplets inside polymersomes (10 mM ATP, 10 mg mL⁻¹ PAH) after lowering the pH condition from 10.5 to pH 7.4. (n=9). b) Plot showing the change in the diameter of the matured coacervate inside these polymersomes with time (n=9).

Surprisingly, we find that changing the direction of pH transition reduces the time scale of overall complex coacervation process to about 8 to 10 min while the size of the matured coacervate droplet remain similar (27.3 ± 1.58 μm) (Supplementary Figure 15). The slower coacervation process for the pH transition from pH 4 to pH 7.4 case is presumably due to the prevalence of positively charged PAH at the initially low pH condition that impedes the maturation into a single droplet due to repulsive forces, as similarly observed by others for poly-L-lysine and ATP.[2] On the other hand, when the pH is lowered from 10.5 to pH 7.4, the amine group of PAH gradually protonate to initiate nucleation with ATP, yielding coacervate droplets with decrease in zeta potential value compared to the case where the pH is increased from pH 4 to pH 7.4.

To further verify this hypothesis, we conducted a separate experiment in which two sets polymersomes each containing i) 10 mM ATP, 5 mg mL⁻¹ PAH, and ii) 5 mM ATP, 10 mg mL⁻¹ PAH, respectively, in the aqueous core were prepared. As the zeta-potential value of the matured coacervate droplet formed in bulk were $+36.8 \pm 4.9$ mV and $+68.0 \pm 2.7$ mV for i) and ii) respectively, inducing coacervation by increasing the pH from 4 to pH 7.4 for i) will show whether less positive surface charge than the control (10 mM ATP, 10 mg mL⁻¹ PAH)

will decrease the time scale of coacervation. Conversely, lowering the pH from 10.5 to pH 7.4 for ii) will reveal whether the time scale will be increased compared to the control by increasing the surface charge.

Supplementary Figure 16. a) Plot showing the change in the number of coacervate droplets inside polymersomes (10 mM ATP and 5 mg mL⁻¹ PAH) after increasing the pH condition from 4 to pH 7.4. (n=10). B) Plot showing the change in the diameter of the matured coacervate inside these polymersomes with time (n=10).

In the first case where the pH was raised from 4 to 7.4 for the polymersomes containing 10 mM ATP and 5 mg mL⁻¹ PAH, we observe that the overall coacervation process took about 35 min and the average size of the resulting matured coacervate droplet is $13.38 \pm 1.15 \mu\text{m}$ (Supplementary Figure 16). This clearly shows that these exhibit faster coacervation compared to that of polymersomes containing 10 mM ATP and 10 mg mL⁻¹ PAH which took more than 1 h in the same direction of pH transition.

Supplementary Figure 17. a) Plot showing the change in the number of coacervate droplets inside polymersomes (5 mM ATP, 10 mg mL⁻¹ PAH) after lowering the pH condition from 10.5 to pH 7.4. (n=9). b) Plot showing the change in the diameter of the matured coacervate inside these polymersomes with time (n=9).

In the second case in which the pH was lowered from 10.5 to 7.4 for the polymersomes containing 5 mM ATP and 10 mg mL⁻¹ PAH, we observe that the PAH and ATP matured into a large coacervate droplet after about 30 min and the average size is $18.73 \pm 2.44 \mu\text{m}$ (Supplementary Figure 17). This confirms that the coacervation time scale has been increased as predicted to exhibit slower coacervation compared to that of polymersomes containing 10 mM ATP and 10 mg mL⁻¹ PAH which took about 10 min to complete in the same direction of pH transition.

Overall, these results reveal that the mechanical process of nucleation and maturation is indeed the rate-determining step in the pH-induced complex coacervation. While the dimension of the polymersome affects the time scale as well as the resulting matured coacervate droplet size by altering the 3D-diffusion length, we also demonstrate that the surface charge of the complex coacervate as well as the direction of pH transition serve as a critical factor. We also note that the coacervation process in similar time scale (about 60 min) has been reported by others using pLys and ATP in lipid-based GUVs upon changing the pH from pH 11 to pH 9.[3]

List of references

- [1] E. A. Frankel, P. C. Bevilacqua & C. D. Keating. Polyamine/Nucleotide Coacervates Provide Strong Compartmentalization of Mg²⁺, Nucleotides, and RNA. *Langmuir* **32**, 2041–2049 (2016)
- [2] K. K. Nakashima, J. F. Baaji & E. Spruijt. Reversible generation of coacervate droplets in an enzymatic network. *Soft Matter* **14**, 361-367 (2018)
- [3] C. Love, J. Steinkühler, D. T. Gonzales, N. Yandrapalli, T. Robinson, R. Dimova & T.-Y. D. Tang. Reversible pH-Responsive Coacervate Formation in Lipid Vesicles Activates Dormant Enzymatic Reactions. *Angew. Chem. Int. Ed.* **59**, 5950-5957 (2020)

These new results appear as **Supplementary Figure 14-17** in the Supplementary Information and the discussions described above have been added in the revised manuscript as follows:

“Moreover, we find that the average diameter of the matured coacervate droplet decreases to $22.5 \pm 3.3 \mu\text{m}$ when the overall solution pH is increased to pH 9.45 above the pKa value of PAH due to the decrease in the charged amine groups in PAH capable of forming complex coacervates with ATP (**Supplementary Fig. 12**). We note that further increasing the pH condition from 7.4 to pH 11 leads to disassembly of the matured coacervate droplet within the polymersome, as evidenced by the transition from a discrete smaller droplet to larger homogeneous fluorescent signal throughout the polymersome (**Supplementary Fig. 13**). While these results demonstrate that complex coacervates can be induced and disassembled inside Pluronic-based polymersomes by altering the external pH condition, we also noticed that the coacervation process is much slower compared to the proton permeation rate.

To verify the origin of this large discrepancy in time scale of proton diffusion and pH-induced complex coacervation, we investigated the effect of overall polymersome size and the electrostatic interaction among PAH and ATP. As the coacervation occurs through nucleation of small coacervate droplets followed by maturation into a single larger one, we first studied how the polymersome size affects the time scale of complex coacervation with the assumption that reducing the size of the polymersome would decrease the time scale since the 3D-diffusion length during maturation process becomes shorter. To achieve this, we prepared sets of polymersomes with identical composition but with different overall sizes and induced complex coacervation by increasing the pH from 4 to 7.4. We observe that the time scale of maturation into a large coacervate droplet decreased by about 20 min and the matured coacervate droplet diameter decreased from 27.4 ± 3.7 to $23.3 \pm 2.0 \mu\text{m}$ when the polymersome size was reduced from 118 to $78.91 \pm 4.54 \mu\text{m}$ (**Supplementary Fig. 14**).

Next, we studied how the electrostatic interaction between PAH and ATP affects the overall coacervation process. Zeta-potential value of the matured coacervate droplet formed in bulk for 10 mM ATP and 10 mg mL^{-1} PAH was $+61.1 \pm 3.8 \text{ mV}$, indicating that there exists excess amount of positively charged PAH compared to ATP with protonated amine groups presented toward the coacervate surface.^[43] To investigate whether the presence of excess amount of protonated PAH initially affects the electrostatic interaction with ATP as well as maturation, we induced complex coacervation in the direction of lowering the pH from pH 10.5 to pH 7.4, as opposed to the direction of increasing the pH from pH 4 to pH 7.4. Unexpectedly, we find that changing the direction of pH transition reduces the time scale of overall complex coacervation process to about 8 to 10 min while the size of the matured coacervate droplet remain similar ($27.3 \pm 1.58 \mu\text{m}$) (**Supplementary Fig. 15**). The slower coacervation process for the pH transition from pH 4 to pH 7.4 case is presumably due to the prevalence of positively charged PAH that impedes the maturation into a single droplet due to repulsive forces, as similarly observed by others for poly-L-lysine and ATP.^[45] On the other hand, when the pH is lowered from 10.5 to pH 7.4, the amine group of PAH gradually protonate to initiate nucleation with ATP, yielding coacervate droplets with decrease in zeta potential value compared to the case where the pH is increased from pH 4 to pH 7.4. Further detailed experimental verifications reveal that the mechanical process of nucleation and maturation is indeed the rate-determining step in the pH-induced complex coacervation (**Supplementary Fig. 15-17**).

While the dimension of the polymersome affects the time scale as well as the resulting matured coacervate droplet size, we also demonstrate that the surface charge of the complex coacervate as well as the direction of pH transition serve as a critical factor. We also note that the coacervation process in similar time scale (about 60 min) has been reported by others using pLys and ATP upon changing the pH from pH 11 to pH 9.^[7]” (page 9, lines 6-34, and page 10, lines 1-18)

• Fig. 3a-b: It is not clear what the fitted lines are: are these polynomial fits or just a guide to the eye? Also, it is quite confusing that the authors have switched the colour codes for graph a and b. Accordingly, the labels for the two sketches in 3b are wrong. Also, the label ‘single phase’ is a bit confusing. Some simple modifications or addition of sketches could make this graph much more understandable.

*We thank the reviewer for the comments. The lines shown in **Fig. 3** (now, **Supplementary Figure S18a** and **S18b**) are simply guide-to-the-eye, and as per reviewer’s suggestion, a brief description is added in the figure caption to improve clarity as follows.*

“Green data points (with dashed line as a guide to the eye)...“

We also thank the reviewer for pointing out this critical typographic error in the figure caption that describes the color code for ATP-PAH and ADP-PAH. We have revised it based on the reviewer’s comment as follows.

“ADP-PAH (green data points) or ATP-PAH (blue data points).“

*We also agree with the reviewer that labeling within the plot such as “single phase” is a bit confusing as the domain is different for each system, ADP-PAH and ATP-PAH, respectively. To avoid such confusion, we removed all labeling but instead put legends in **Supplementary Figure S18a** and **S18b** to clearly distinguish among two systems.*

• The authors have done a systematic and a detailed study of how ATP versus ADP is a better coacervation agent (section 2.3). However, this is a quite well-known fact in the field and in my opinion, not really a new addition. However, this is more of a side-remark, and I leave it up to the authors to modify that section or keep it as it is.

*We much appreciate the reviewer’s constructive comments and we agree that the study of coacervation of polycations with either ATP or ADP is well-known [1-3]. However, at the same time, finding the optimal condition at which ADP does not form coacervate with PAH while equivalent amount of ATP readily forms coacervate with PAH is the key conceptual basis for the demonstration of PEP-driven enzymatic conversion of ADP to ATP that leads to complex coacervation in polymersomes. Therefore, we have reduced this part in the revised manuscript and relocated previous **Fig. 3** along with the details to Supplementary Information.*

List of references

[1] E. A. Frankel, P. C. Bevilacqua & C. D. Keating. Polyamine/Nucleotide Coacervates Provide Strong Compartmentalization of Mg²⁺, Nucleotides, and RNA. *Langmuir* **32**, 2041–2049 (2016)

[2] K. K. Nakashima, J. F. Baaji & E. Spruijt. Reversible generation of coacervate droplets in an enzymatic network. *Soft Matter* **14**, 361-367 (2018)

[3] I. B. A. Smokers, M. H. I. van Haren, T. Lu & E. Spruijt. Complex coacervation and compartmentalized conversion of prebiotically relevant metabolites. *ChemSystemsChem* e202200004 (2022)

- It will be interesting to see if the formed coacervates can be dissolved by lowering the pH or through enzymatic means and will be a worthy addition to the manuscript.

*We thank the reviewer for the helpful suggestions, and we agree that demonstrating the disassembly of coacervate by changing the pH condition would be a worthy addition to the manuscript. While pH-induced disassembly can be theoretically executed in both directions, either by lowering the pH or increasing the pH, the zeta-potential value of the matured coacervate droplet for our system is $+61.1 \pm 3.8$ mV, as discussed previously. This indicates that there exists excess amount of positively charged PAH compared to ATP with protonated amine groups ($pK_a \sim 9$) presented toward the coacervate surface.[1] Therefore, we chose to demonstrate the reversibility of the coacervate droplets through deprotonating the PAH by increasing the pH condition above the pK_a value. We observe that increasing the pH condition from 7.4 to pH 11 leads to disassembly of the matured coacervate droplet within the polymersome, as evidenced by the transition from a discrete smaller droplet to larger homogeneous fluorescent signal throughout the polymersome shown in **Supplementary Figure 13**. This indicates that the complex coacervation can be not only induced but also disassembled by altering the external pH condition.*

Supplementary Figure 13. Fluorescence micrographs of polymersomes (10 mM ATP and 10 mg mL⁻¹ PAH) before and after increasing the pH condition from 7.4 to pH 11. Scale bar represents 100 μ m.

*As per reviewer' comment, we have added the newly prepared **Supplementary Figure 13** in the Supplementary Information and added a brief description in the revised manuscript as follows:*

“We note that further increasing the pH condition from 7.4 to pH 11 leads to disassembly of the matured coacervate droplet within the polymersome, as evidenced by the transition from a discrete smaller droplet to larger homogeneous fluorescent signal throughout the polymersome (Supplementary Fig. 13). While these results demonstrate that complex coacervates can be induced and disassembled inside Pluronic-based polymersomes by altering the external pH condition, we also noticed that the coacervation process is much slower compared to the proton permeation rate.” (page 9, lines 9-16)

List of references

[1] E. A. Frankel, P. C. Bevilacqua & C. D. Keating. Polyamine/Nucleotide Coacervates Provide Strong Compartmentalization of Mg²⁺, Nucleotides, and RNA. *Langmuir* **32**, 2041–2049 (2016)

Minor points

- The first sentence of the abstract: the part mentioning the cytoskeleton needs revising.

We thank the reviewer for the suggestion, and we agree that the first sentence of the abstract may be misleading. As per reviewer’s suggestion, we have revised the first sentence of the abstract as follows:

“Living cells can spatiotemporally control biochemical reactions to dynamically assemble membraneless organelles and remodel cytoskeleton.”

- Fig. 4b: the polymersomes can hardly be seen, please increase the contrast.

*We thank the reviewers for comments. As per the reviewer’s comment, we enhanced the color contrast to clearly visualize the polymersomes shown in **Revised Fig. 3b**.*

- Fig. 5b: the images are very faint. Also, it is hard to make out the different coloured circles in the graph.

*We thank the reviewers for the suggestion. We have enhanced the color contrast for the symbols as well as the fluorescence micrographs of **Revised Fig. 4b** to clearly distinguish them.*

REVIEWER COMMENTS

Reviewer #1 (Remarks to the Author):

The authors have addressed all my comments and the manuscript has been significantly improved. I have no objection for the publication of this work as is.

Reviewer #2 (Remarks to the Author):

This is a second review on the study "Microfluidic approach of artificial cell-like polymersomes to spatiotemporally control the signal-driven enzymatic reactions". Authors postulate that there are limited approaches for low-molecular weight substrates diffusion through membrane vesicles without the help of biopores etc. Especially, for the design and fabrication of giant unilamellar vesicles (GUV) by microfluidic approaches. This is true when only considering this specific technology for the design and fabrication of synthetic, polymeric GUVs. There is a need of further improvements for the design and fabrication of spatiotemporal control over enzymatic (cascade) reactions, while other approaches in the literature of controlled permeability of proteinosomes, polymersomes and membrane-stabilized coacervates etc. are still existing, leading to with and without integrated biopores, membrane proteins and channel proteins for specific molecule diffusions for carrying out spatiotemporal control over enzymatic reactions.

Authors have further deeply replied on all concerns of reviewer's comments and revised the manuscript as well as the supporting information, while the conclusion/discussion part has not been obviously improved in respect to other cell-like structures.

Generally, this is a very impressive work with some new aspects, while the novelty of this work is further limited due a very limited exchange of information (ions and substrates) for crossing the semipermeable membrane of polymersomes, produced by the use of microfluidic approach. When considering the very limited membrane permeability which is of high interest to exchange also larger biomolecules or biomacromolecules, then a limited novelty is given at which spatiotemporal control of cell communication can be carried out on a lower level. It is really pity that no ATP can be delivered afterwards to the lumen of polymersomes in this study. This means for future use that all larger components have to be encapsulated first, while very low molecular-weight components can be added later for crossing semi-permeable polymersomes membrane. Thus one imagines that no additional diffusion of components from outside to inside will happen when molar masses between 1-10kDA exist.

The term of spatiotemporal control is used in a very simplified case, when no exchange between two different compartments may occur, when larger intermediates are produced which are needed in the second compartment for the establishment of final product. Thus, a limited use of this potential platform can be foreseen.

Reviewer #3 (Remarks to the Author):

I am satisfied with the detailed revision of the manuscript and I have one last suggestion. Regarding my first question on the dewetting, the authors have clearly explained how the pocket remains attached to the polymersome in their rebuttal. However, they do not explicitly say that in the main text. I think it is important to mention this in the main text to avoid any ambiguity and to make it clear that a small solvent pocket remains attached to the polymersome.

Once authors clarify this additional detail in the main text, I recommend publication.

REVIEWER COMMENTS

Reviewer #1 (Remarks to the Author):

The authors have addressed all my comments and the manuscript has been significantly improved. I have no objection for the publication of this work as is.

We much appreciate the reviewer's positive evaluations. We are delighted to see that the revision that we provided is well-delivered and that the reviewer has no objection for the publication of this work as it is.

Reviewer #2 (Remarks to the Author):

This is a second review on the study “Microfluidic approach of artificial cell-like polymersomes to spatiotemporally control the signal-driven enzymatic reactions”. Authors postulate that there are limited approaches for low-molecular weight substrates diffusion through membrane vesicles without the help of biopores etc. Especially, for the design and fabrication of giant unilamellar vesicles (GUV) by microfluidic approaches. This is true when only considering this specific technology for the design and fabrication of synthetic, polymeric GUVs. There is a need of further improvements for the design and fabrication of spatiotemporal control over enzymatic (cascade) reactions, while other approaches in the literature of controlled permeability of proteinosomes, polymersomes and membrane-stabilized coacervates etc. are still existing, leading to with and without integrated biopores, membrane proteins and channel proteins for specific molecule diffusions for carrying out spatiotemporal control over enzymatic reactions.

Authors have further deeply replied on all concerns of reviewer's comments and revised the manuscript as well as the supporting information, while the conclusion/discussion part has not been obviously improved in respect to other cell-like structures. Generally, this is a very impressive work with some new aspects, while the novelty of this work is further limited due a very limited exchange of information (ions and substrates) for crossing the semipermeable membrane of polymersomes, produced by the use of microfluidic approach. When considering the very limited membrane permeability which is of high interest to exchange also larger biomolecules or biomacromolecules, then a limited novelty is given at which spatiotemporal control of cell communication can be carried out on a lower level. It is really pity that no ATP can be delivered afterwards to the lumen of polymersomes in this study. This means for future use that all larger components have to be encapsulated first, while very low molecular-weight components can be added later for crossing semi-permeable polymersomes membrane. Thus one imagines that no additional diffusion of components from outside to inside will happen when molar masses between 1-10kDA exist. The term of spatiotemporal control is used in a very simplified case, when no exchange between two different compartments may occur, when larger intermediates are produced which are needed in the second compartment for the establishment of final product. Thus, a limited use of this potential platform can be foreseen.

We appreciate the reviewer's constructive criticism and agree that the conclusion section has not been considerably improved in the previous version of the revised manuscript in respect to other cell-like structures. In fact, we only compared the stability of Pluronic-based polymersomes with lipid-based GUVs which is insufficient as functionality (e.g. tunable membrane permeability) is also the key determinant of polymersomes.

As per reviewer's suggestion, we have performed a new experiment in which we prepared two sets of analogous polymersomes with comparable size but with different membrane composition to demonstrate that the membrane permeability can be altered to precisely regulate small substrates without the use of biopores. In the first set of polymersomes, we replaced a portion of Pluronic L121 with Pluronic L61 (PEO₂-PPO₃₀-PEO₂, MW ~ 2000) which exhibit similar f value but smaller molecular weight (MW) than Pluronic L121 (MW ~ 4400) by preparing a mixture of 75 mol% Pluronic L121 and 25 mol% Pluronic L61 with a total of 20 wt% polymer dissolved in a mixture of chloroform and cyclohexane (36:64 vol%). This is anticipated to reduce the membrane thickness and thus increase the membrane permeability, as similarly reported by others. [1] For the other set of polymersomes, we used poly(butadiene)-*b*-poly(ethylene oxide)(PB-PEO) block copolymer as the membrane constituent, which has been reported to exhibit higher f value (~ 0.4), [2] and result in a thicker membrane. [3,4] Then, we compared the normalized fluorescence intensity change within these sets of polymersomes over time for two fluorescent dye molecules with MW near 500 Da, rhodamine 6G (479 Da) and HPTS (524 Da), respectively, as shown in the newly added **Supplementary Figure S27** below.

(Newly added) Supplementary Figure 27. (a) Plot showing the normalized fluorescence intensity change over time within the three sets of polymersomes with different compositions using Rhodamine 6G (R6G). Blue dot refers to PB-PEO polymersome, while the red and green dot refers to Pluronic L121-based polymersome (control) and Pluronic L121 + L61 (75:25 mol%) blended polymersome. (b) Plot showing the normalized fluorescence intensity change over time within the sets of polymersomes with different compositions using HPTS. The dotted lines in both plots represent the model fit used to determine the membrane permeability for each diffusing species. (c) Plot showing the estimated permeability of two fluorescent dye molecules with respect to different membrane compositions in polymersomes.

We observed that the permeability of R6G (479 Da) in Pluronic L121+L61 blended polymersomes is slightly higher (6.93 nm/s) than the analogous Pluronic L121 only polymersomes (6.28 nm/s). Moreover, we found that the PB-PEO polymersomes with higher f

value and a thicker membrane exhibit low molecular permeability (0.025 nm/s), which was even lower than the permeability of HPTS in Pluronic L121 polymersomes. Furthermore, similar tendency was observed for HPTS (524 Da) in which we observed increase in HPTS permeability by approximately two-folds from 0.051 to 0.096 nm/s by blending with Pluronic L61 and extremely low permeability for PB-PEO polymersomes (0.016 nm/s). Overall, these results clearly indicate that the membrane permeability can be either increased or decreased by mixing with lower molecular weight analogues or utilizing higher f value polymer without using biopores.

In fact, the main design principle of this paper is to prepare semi-permeable polymersomes that are size-selective, allowing low molecular weight ions, phosphoenol pyruvate (PEP), and adenosine diphosphate (ADP) to readily diffuse through the membrane while adenosine triphosphate (ATP) and pyruvate kinase (PyK) cannot, to exploit the external signal-driven complex coacervate formation in polymersomes via *in vitro* enzymatic reactions.

While we also agree with the reviewer that lack of permeability to components that exhibit molar masses above 1 kDa may limit the use of this potential platform, we believe that this is beyond the scope of this paper as this is achievable in polymersomes by simply incorporating biopores into the membrane, even though they are expensive, difficult to purify, and insert desired quantity into the membrane. In fact, various polymersomes have been shown to successfully incorporate membrane proteins in their block copolymer membrane to transport biomolecules.[5-7] Therefore, we believe that our biopore-free polymersomes offer unique set of advantages including size controllability, efficient encapsulation, enhanced stability, and tunable molecular permeability by either changing the membrane constituent or inserting biopores, all of which clearly demonstrates the potential of this artificial cell-like systems in spatiotemporally controlling complex biochemical reaction pathways and cascade reactions.

Nevertheless, we thank the reviewer for the helpful comments and suggestions, which have substantially improved the clarity of our manuscript. We also hope that with the additional experiments and revisions that we are providing, the reviewer will now agree with the other reviewers who stated that we have addressed all the comments and recommend publication of this work.

Based on the reviewer's comment, we have extended the discussion section of the revised manuscript as follows with the experimental details included in the Supporting Information due to the main text wording limit:

“As the strategy outline in this work is applicable to broad range of amphiphilic block copolymers due to advances in controlled polymer synthesis, the membrane composition can be fine-tuned to control the permeability without the use of biopores. Indeed, preparing analogous sets of polymersomes with different compositions (Pluronic L121 and L61 blended polymersomes and poly(butadiene)-b-poly(ethylene oxide) (PB-PEO) polymersomes) and comparing the fluorescence intensity change with Pluronic L121-based polymersomes reveal that the membrane permeability can be either increased or decreased by mixing with lower molecular weight analogues or utilizing high f value polymer. (See the details in **Supporting Information Figure S27 and Notes**). However, we note that in case where exchange of larger biomolecules with molar masses above 1 kDa is needed between two different compartments for the spatiotemporal control of biochemical reactions, membrane proteins can be incorporated into the membrane to transport these biomolecules, as reported previously by others [6-8]. Furthermore, since the model reactions demonstrated in this work can be further extended to...” (page 19, lines 1-13)

List of references

- [1] do Nascimento, D. F., Arriaga, L. R., Eggersdorfer, M., Ziblat, R., Marques, M. de F., Reynaud, F., Koehler, S. A. & Weitz, D. A. Microfluidic Fabrication of Pluronic Vesicles with Controlled Permeability. *Langmuir* **32**, 5350-5355 (2016).
- [2] Habel, J., Ogbonna, A, Larsen, N., Cherré, S., Kynde, S., Midtgaard, S. R., Kinoshita, K., Krabbe, S., Jensen, G. V., Hansen, J. S., Almdal, K. & Hélix-Nielsen, C. Selecting analytical tools for characterization of polymersomes in aqueous solution. *RSC Adv.* **5**, 79924-79946 (2015).
- [3] Lim, S. K., De Hoog, H.-P., Parikh, A. N., Nallani, M. & Liedberg, B. Hybrid, Nanoscale Phospholipid/Block Copolymer Vesicles. *Polymers* **5**, 1102-1114 (2013).
- [4] Seo, H., Nam, C., Kim, E., Son, J. & Lee, H. Aqueous Two-Phase System (ATPS)-Based Polymersomes for Particle Isolation and Separation. *ACS Appl. Mater. Interfaces* **12**, 55467-55475 (2020).
- [5] Hutchison, J. M., Poust, S. K., Kumar, M., Cropek, D. M., MacAllister, I. E., Arnett, C. M. & Zilles, J. L. Perchlorate reduction using free and encapsulated *Azospira oryzae* enzymes, *Environ. Sci. Technol.* **47**, 9934 (2013).
- [6] Belluati, A., Thamboo, S., Najer, A., Maffei, V., von Planta, C., Craciun, I., Palivan, C. G. & Meier, W. Multicompartment polymer vesicles with artificial organelles for signal-triggered cascade reactions including cytoskeleton formation. *Adv. Funct. Mater.* **30**, 2002949 (2020).
- [7] Kleineberg, C., Wölfer, C., Abbasnia, A., Pischel, D., Bednarz, C., Ivanov, I., Heitkamp, T., Börsch, M., Sundmacher, K. & Vidaković-Koch, T. Light-driven ATP regeneration in diblock-grafted hybrid vesicles. *ChemBioChem* **21**, 2149–2160 (2020).

Reviewer #3 (Remarks to the Author):

I am satisfied with the detailed revision of the manuscript and I have one last suggestion. Regarding my first question on the dewetting, the authors have clearly explained how the pocket remains attached to the polymersome in their rebuttal. However, they do not explicitly say that in the main text. I think it is important to mention this in the main text to avoid any ambiguity and to make it clear that a small solvent pocket remains attached to the polymersome.

Once authors clarify this additional detail in the main text, I recommend publication.

We thank the reviewer for acknowledging that we have adequately revised the manuscript. As for the suggestion, we have included a brief sentence in the revised manuscript to explicitly mention that the pocket remains attached to the polymersome as follows:

“We note that minuscule oil pockets remain attached to the polymersomes but eventually thin out over time and do not affect the experimental results.”(page 5, lines 9-11)

REVIEWERS' COMMENTS

Reviewer #2 (Remarks to the Author):

The reviewer's last open points were smoothly considered and replied. I strongly recommend this manuscript for publishing.

Many thanks for considering this impressive work!